# ENERGY SHIELDS FOR FAIRNESS

## ABSTRACT

Runtime fairness is not a one-time constraint but a dynamic property evaluated over a sequence of decisions. To ensure fairness at runtime it is necessary to account for past decisions, information neglected by conventional, static classifiers. Traditional fairness shields enforce runtime fairness abruptly, by intervening *deterministically* whenever a sequence of decisions violates the target for a running fairness measure. This motivates our *main conceptual contribution: **energy shields***. An energy shield is a novel, lightweight, adaptive controller that monitors a sequence of decisions and intervenes *probabilistically* to ensure runtime fairness smoothly, by utilizing physics-inspired energy functions to nudge the sequence towards fairness: the more unfair the decisions, the stronger the nudging force becomes. This makes energy shields the ***first*** fairness shields to provide both *short-term safety and long-term liveness guarantees*. Safety ensures that the running fairness measure stays within a running target interval with high probability, and liveness ensures that the limit of the fairness measure lies within the limit target interval. Intuitively, the short-term specifies the tolerated fairness values and the long-term specifies the desired fairness values. We also provide a synthesis procedure for constructing the least intrusive energy shield for a given target specification, and demonstrate its efficiency experimentally. As a sanity check for the theoretical contributions, we evaluate our energy shields against existing fairness shields through the lens of short- and long-term fairness.

## 1 INTRODUCTION

Algorithmic decision making is ubiquitous in modern life, from hiring and lending to online advertising. In these settings, binary decisions, e.g., approving or denying a loan or displaying one ad over another, are often made sequentially Liu et al. (2018); D'Amour et al. (2020). Much research has focused on designing *statically fair* algorithms, which ensure fairness in expectation over a fixed distribution Caton & Haas (2020). This guarantees that in the long term the sequence of decisions will be fair (as long as the distribution does not shift). However, this perspective on fairness fails to address unfair behavior that arises in the short term from the cumulative history of decisions. In particular, a statically fair classifier may produce arbitrarily biased sequences in the short term Cano et al. (2025a). For illustration, we use a simplified binary online advertisement setting.

**Example 1.** *Two companies, A and B, bid $c_A$ and $c_B$ for ad space. At each time $t \in \mathbb{N}$, a user sees an ad from A ($X_t = 1$) with decision probability $p$, or from B ($X_t = 0$) otherwise. In the static setting, the decision maker is fair if $p$ matches the bid ratio, i.e., $p \in c_A/(c_A + c_B) \pm \varepsilon$ for $\varepsilon > 0$. In the sequential setting, a sequence $x_1, \ldots, x_t$ of ad placements is fair at time $t$ if the empirical running average of ads matches the bid ratio, i.e., $\mu(x_1, \ldots, x_t) = \frac{1}{t}\sum_{i=1}^{t} x_i \in c_A/(c_A + c_B) \pm \varepsilon$ for $\varepsilon > 0$. We want $\mu(x_1, \ldots, x_t)$ to be always in the interval. Furthermore, the sequence of ad placements is fair in the limit if $\lim_{t \to \infty} \mu(x_1, \ldots, x_t) = p$.*

**Running vs. limit fairness: safety and liveness.** Ex. 1 illustrates two *fairness* properties defined w.r.t. a running and a limit target interval. The short-term property requires the running target interval to be met by the running average at a finite time, and the long-term property requires the limit target to be met by the limit average of the decision sequence. Intuitively, the running target expresses what fairness values we tolerate in the short-term, and the limit target expresses what fairness values we desire in the long-term. This distinction matches safety-liveness classification of properties studied in formal verification Baier & Katoen (2008). The violation of a safety property can be determined at

a finite time, thus matching our short-term requirement for fairness, which we call *running fairness*. The satisfaction of a liveness property can be determined in the limit only, thus matching our long-term requirement for fairness, which we call *limit fairness*. Our objective is to ensure that a decision sequence meets given target intervals for both running and limit fairness.

**Traditional vs. energy shielding.**   To enforce dynamic fairness requirements, *runtime shielding* has emerged as a promising approach Cano et al. (2025b). Originating in formal verification, a shield is a lightweight controller that monitors a decision sequence and intervenes minimally to correct decisions in order to enforce desired fairness requirements Alshiekh et al. (2018). Unfortunately, existing fairness shields act deterministically, alternating abruptly between full intervention and no intervention, to enforce highly restrictive short-term fairness requirements, and they fail to provide long-term fairness guarantees altogether Cano et al. (2025b). Here, we introduce gentle, probabilistic shielding mechanisms that have both short- and long-term fairness guarantees.

**Contributions.**   Our main *conceptual contribution* is the introduction of ***energy shields***, a probabilistic shielding framework inspired by physics-based energy functions to ensure both running and limit fairness requirements. An energy shield smoothly nudges a sequence of decisions toward fairness by assigning higher "energy" to unfair sequences and intervening, proportional to the energy, to enter a lower-energy state. As a consequence, our *energy shields are the **first** fairness shields capable of providing short- and long-term fairness guarantees* To demonstrate that energy shields provide **short-term fairness** guarantees, we provide exponentially decaying tail bounds on the probability and expected value of running fairness violations with respect to a given target interval. To demonstrate that energy shields provide **long-term fairness** guarantees, we characterize the limit behavior of the shielded process, deriving conditions on the energy function to ensure convergence of the fairness measure to a given target. Additionally, we quantify the long-run expected cost of intervention and prove that steeper energy functions yield fewer violations, an important monotonicity property. We exploit the monotonicity property of energy functions to propose a **synthesis procedure** that combines binary search and dynamic programming with tail bounds to find the least intrusive energy shield satisfying the desired running and limit fairness properties. We validate the effectiveness and efficiency of the synthesis procedure experimentally. Moreover, we benchmark our energy shields against the seminal energy shields by Cano et al. (2025b), empirically supporting the claim that energy shields are the first shields to provide both short- and long-term guarantees.

As the bulk of our contribution is theoretical, we focus in the main text on presenting results and their intuitions. The complete proofs of all statements are provided in the appendix.

## 2 RELATED WORK

Most algorithmic fairness research focuses on fairness in a static setting. This includes: measures for the fairness of a decision maker at the level of groups Feldman et al. (2015); Hardt et al. (2016) and individuals Dwork et al. (2012); pre-, in-, and post-processing techniques to synthesize fair decision maker Hardt et al. (2016); Gordaliza et al. (2019); Zafar et al. (2019); Agarwal et al. (2018); Wen et al. (2021); verification techniques to check whether static decision makers are fair Albarghouthi et al. (2017); Bastani et al. (2019); Sun et al. (2021); Ghosh et al. (2021); Meyer et al. (2021); Li et al. (2023). Among the existing techniques for static fairness, our shields could be classified as a post-processing technique. The key difference is that those methods modify decision-makers *once* before deployment, whereas our work addresses fairness *during* deployment via runtime intervention.

We are not the first to be concerned with algorithmic fairness over a sequence of decisions Alamdari et al. (2024); Cano et al. (2025a). A large body of work focuses on detecting unfair behavior at runtime, both for individual Gupta et al. (2025) and for group fairness Albarghouthi & Vinitsky (2019); Henzinger et al. (2023a); Baumeister et al. (2025). Beyond detection, Cano et al. (2025b) is the only work enforcing fairness at runtime. Their shields adopt the sequential fairness definition from Parand et al. Alamdari et al. (2024), ensuring that a sequence of decisions will be fair with probability 1 at predefined periodic intervals. Our shields soften this condition, providing high-probability short-term guarantees and are the first to provide limit guarantees.

Although the monitoring and enforcement of fairness has only recently emerged as a topic of interest, classical runtime monitoring and enforcement have long been studied in the runtime verification

community, with monitors Stoller et al. (2011); Faymonville et al. (2017); Maler & Nickovic (2004); Donzé & Maler (2010); Bartocci et al. (2018); Baier et al. (2003) and shields Carr et al. (2023); Al-shiekh et al. (2018); Córdoba et al. (2023) developed for Linear Temporal Logic specifications. Shielding has also been explored in probabilistic settings Jansen et al. (2020); Yang et al. (2023), drawing on techniques from probabilistic model checking Katoen (2016). We build on results from stochastic approximation Borkar (2008); Karandikar & Vidyasagar (2024) to design gentler probabilistic shields that preserve almost sure convergence guarantees.

## 3 SETTING

In this section, we formally introduce the setting, the fairness properties, and the notion of a shield.

**Decision process.** We model the setting in Ex. 1 using a single coin with decision probability $p \in [0, 1]$. At each point in time $t \in \mathbb{N}$ the coin is tossed, resulting in an decision $x_t \in \mathbb{B}$, where $\mathbb{B} = \{0, 1\}$, which is the realization of the random variable $X_t \sim \text{Bernoulli}(p)$. Combined they generate the decision process $X = (X_t)_{t \in \mathbb{N}}$ consisting of i.i.d. Bernoulli $p$ random variables. A realization $x = (x_t)_{t \in \mathbb{N}} \in \{0, 1\}^\omega$ of $X$ is an infinite sequence of binary values.

**Fairness.** We are interested in measuring the fairness of the process defined above. The measure we use is called average outcome fairness Cano et al. (2025a). Formally, given an infinite sequence of binary decisions $x \in \{0, 1\}^\omega$, e.g., a realization of the decision process, we measure the fairness of a finite prefix $x_{1:t} = (x_1, \ldots, x_t)$ as its average decision, denoted by $\mu(x_{1:t})$, or $\mu_t$ if clear from the context. We use $\mu(x)$ to denote the fairness measure of the realization in the limit, if it exists:

$$\mu(x_{1:t}) = \frac{1}{t} \sum_{i=1}^{t} x_i, \qquad \text{and} \qquad \mu(x) = \lim_{t \to \infty} \mu(x_{1:t}). \qquad (1)$$

A *fairness target* is a tuple $\varphi = (\tau, \mathcal{S}, \mathcal{L})$ consisting of a burn-in time $\tau \in \mathbb{N}$, a running target $\mathcal{S} \subseteq [0, 1]$, and a limit target $\mathcal{L} \subseteq [0, 1]$. The fairness target specifies the acceptable fairness measure values at every finite time greater than the burn-in $\tau$ and at the limit. For the target to be satisfiable we require that the running and limit target intersect $\mathcal{S} \cap \mathcal{L}$ and that the burn-in is sufficiently large to account for initial high variance. Given a fairness target, a infinite sequence $x \in \mathbb{B}^\omega$ satisfies:

- *point fairness* at time $t \geq \mathbb{N}$, if the fairness measure is in the running target at $t$, i.e., $\mu(x_{1:t}) \in \mathcal{S}$;
- *running fairness*, if point fairness is always satisfied after the burn-in $\tau$, i.e., $\forall t \geq \tau \colon \mu(x_{1:t}) \in \mathcal{S}$;
- *limit fairness*, if the fairness measure is in the limit target in the limit, i.e., $\lim_{t \to \infty} \mu(x_{1:t}) \in \mathcal{L}$;
- *fairness*, if both running fairness and limit fairness are satisfied.

**Example 2** (Ex. 1 cont.). *Assume the bid ratio is* 0.5. *In the long-term we require a fairness measure of* 0.5, *and in the short-term we accept a tolerance of* 0.1 *after some burn-in* $\tau$. *The corresponding fairness target is* $\varphi = (\tau, [0.4, 0.6]\{0.5\})$. *Note, the decision* $X_t$ *follows a Bernoulli distribution, thus the fairness measure* $\mu(X_{1:t})$ *follows a binomial distribution scaled by* $1/t$. *Since* $(1/t)Bin(t, p) \xrightarrow{t \to \infty} p$, *limit fairness requires* $p = 0.5$. *The probability of satisfying point fairness at* $t$ *is* $\mathbb{P}[\mu(X_{1:t}) \in \mathcal{S}] = \mathbb{P}[Bin(t, 0.5) \in t(0.5 \pm 0.1)] \approx \sum_{i=\lfloor 0.4t \rfloor}^{\lceil 0.6t \rceil} \binom{t}{i} p^i (1-p)^{t-i}$.

**Shielding.** As illustrated in Ex. 2, without control of $p$, the only possible intervention on the process is to overwrite individual decisions. This is called shielding. A deterministic shield for a decision process is a program with the power to flip the decisions made at runtime. Formally, a deterministic shield $\mathfrak{S} \colon (\mathbb{B} \times \mathbb{B})^* \times \mathbb{B} \to \mathbb{B}$ uses the history of decisions $x_1, \ldots, x_t \in \mathbb{B}^t$ at time $t \in \mathbb{N}$ and the history of intervention $y_1, \ldots, y_{t-1} \in \mathbb{B}^{t-1}$ to compute the next intervention $y_t$. The intervention indicates whether the decision $x_t$ is flipped, i.e., if $y_t = 1$ then the shielded decision is $z_t = 1 - x_t$, otherwise the shielded decision is $z_t = x_t$. The objective of the shield is to aid the satisfaction of the fairness target, evaluated over the sequence shielded decisions $z = (z_t)_{t \in \mathbb{N}}$ with as little interference as possible. To measure this, we define the average interference cost $\nu_t$ over a sequence of interventions $y_1, \ldots, y_t$ as the average number of interventions $\nu_t = (1/t) \sum_{i=1}^{t} y_i$.

**Example 3** (Ex. 2 cont.). *A trivial shield guaranteeing running and limit fairness, ensures that the company ads alternate, i.e., the shielded decision sequence is* $(01)^\omega$, *thus* $\mu(z_{1:2t}) = 0.5$ *for all* $t$.

*A less restrictive shield that satisfies running fairness, but not the limit fairness, is one that only interferes if the point fairness is about to be violated. We call this shield the* naive shield. *For example, assume at $t = 100$ we have $\mu(z_{1:t}) = 40/100$, if $x_{t+1} = 0$, then the fairness measure would be $40/101 < 0.4$, so the shield enforce enforces, i.e., $z_{t+1} = 1$.*

**Probabilistic shields.**   The deterministic shields act aggressively and apruptly once the fairness measure is at risk of leaving the target and remains idle most of the time. This leads to two regimes, one where the shield has full control over the decision and one where it has none. We introduce probabilistic shields where the intervention is done probabilistically, allowing for a much more gentle approach to shielding. Formally, a probabilistic shield is a function $\mathfrak{S}\colon (\mathbb{B} \times \mathbb{B})^* \times \mathbb{B} \to [0, 1]$ mapping a history of decisions $x_1, \ldots, x_t \in \mathbb{B}^t$ at time $t \in \mathbb{N}$ and the history of interventions $(y_1, \ldots, y_{t-1})$ into an nudging probability $q_t = \mathfrak{S}(x_1, y_1, \ldots, y_{t-1}, x_t)$, defining the distribution from which the next intervention is sampled, i.e., $Y_t \sim \text{Bernoulli}(q_t)$.

**Problem sketch.**   We consider four processes: the decision process $X = (X_t)_{t \in \mathbb{N}}$, the intervention process $Y = (Y_t)_{t \in \mathbb{N}}$, the shielded decision process $Z = (Z_t)_{t \in \mathbb{N}}$, and the shielded fairness process $M = (M_t)_{t \in \mathbb{N}}$. The processes are defined at every time $t \in \mathbb{N}$ as follows:

$$X_t \sim \text{Bernoulli}(p), \quad Y_t \sim \text{Bernoulli}(\mathfrak{S}(X_1, Y_1, \ldots, Y_{t-1}, X_t)),$$
$$Z_t = (1 - X_t) \cdot Y_t + X_t \cdot (1 - Y_t), \quad \text{and} \quad M_t = \mu(Z_1, \ldots, Z_t).$$

Intuitively, the shielded fairness process, i.e., the fairness measure evaluated over the shielded decision process, should satisfy a given fairness target $\varphi = (\tau, \mathcal{S}, \mathcal{L})$ with a high-probability, i.e.,

$$\mathbb{P}(\forall t \geq \tau\colon M_t \in \mathcal{S}) \geq 1 - \delta, \quad \text{and} \quad \mathbb{P}(\lim_{t \to \infty} M_t \in \mathcal{L}) \geq 1 - \delta \quad \text{for } \delta \in (0, 1).$$

## 4   Energy-based Shields

We introduce energy shields, a family of physics inspired probabilistic shields. The shields acts by computing an energy state for the process and nudging the system toward lower-energy configurations. The energy state is given by an energy function and the current fairness measure.

**Energy function.**   An energy function is bowl-shaped with minimum at its pivot point.

**Definition 1.** An *energy function* with *pivoting point* $\kappa \in [0, 1]$ is a function $\zeta\colon [0, 1] \to [0, 1]$ satisfying: (i) $\zeta$ is continuously differentiable, i.e., $\zeta$ is differentiable, and $\zeta'$ is continuous; (ii) $\zeta'(c) \leq 0$ for $c < \kappa$, and $\zeta'(c) \geq 0$ for $c > \kappa$; (iii) $\zeta(\kappa) = \zeta'(\kappa) = 0$, and $\zeta(c) > 0$ for $c \in \{0, 1\}$.

**Example 4.** *Two families of energy functions are even-polynomials and exponential energy functions (see Fig. 1a) defined, as $\zeta^{\text{Pol}}_{\kappa,\alpha,\beta}(x) = \alpha|x - \kappa|^\beta$ and $\zeta^{\text{Exp}}_{\kappa,\rho,\sigma}(x) = \rho(1 - e^{-\sigma(x-\kappa)^2})$, where $\kappa \in (0, 1)$, $\beta \in (1, \infty)$, $\alpha \in (0, 1/\max(\kappa, 1-\kappa)^\beta)$, $\sigma \in (0, \infty)$, and $\rho \in (0, 1/(1 - e^{-\sigma(\min\{\kappa, 1-\kappa\})^2}))$. The parameter ranges ensure that the energy function does not exceed $1$ without clipping.*

**Energy shield.**   An energy shield is defined w.r.t. an energy function $\zeta\colon [0, 1] \to [0, 1]$ with pivoting point $\kappa \in [0, 1]$. The pivot point determines the favored decision, while the energy function determines the nudging probability. Formally, assume we have observed the decisions $x_1, \ldots, x_t$ and accumulated the interventions $y_1, \ldots, y_{t-1}$, which determined the shielded decisions $z_1, \ldots, z_{t-1}$ at time $t \in \mathbb{N}$. Then if $\mu_{t-1} \leq \kappa$, the shield accepts $x_t = 1$ and flips $x_t = 0$ with probability $\zeta(\mu_{t-1})$, and if $\mu_{t-1} > \kappa$, the shield accepts $x_t = 0$, and flips $x_t = 1$ with probability. This determines the distribution over the next shielded decision $Z_t$ and fairness value $M_t$.

**Claim 1** (Shielded decision process). *A decision process $Z$ shielded by an energy shield forms a sequence of Bernoulli random variables with evolving bias, i.e, $Z_t \sim \text{Bernoulli}(p_t)$. The biases are defined recursively as $p_1 = 1$ and $p_{t+1} = f(\mu_t)$ for a given history $z_1, \ldots, z_t$, where*

$$f(\mu) = \begin{cases} p + (1-p)\zeta(\mu) & \text{if } \mu \leq \kappa, \\ p \cdot (1 - \zeta(\mu)) & \text{if } \mu > \kappa \end{cases}. \tag{2}$$

*Moreover, the resulting shielded fairness process update can be written as*

$$M_t = \mu_{t-1} + \frac{1}{t}(Z_t - \mu_{t-1}) \quad \text{(with } \mu_0 = 0). \tag{3}$$

Intuitively, Eq. 2 shows that the decision maker pulls the fairness value $\mu$ toward $p$, while the shield exerts an opposing pull toward $\kappa$. Stronger energy values amplify this effect, shifting the process further to $\kappa$. Eq. 3 makes this dynamic explicit: the sequence of $\mu_t$'s evolves as a stochastic approximation process drifting in the direction of $\mathbb{E}[Z_{t+1}|\mu_t] - \mu_t = f(\mu_t) - \mu_t$. At a convergence point, this drift should vanish, which occurs at a fixed point of $f$, i.e., a value of $\mu$ satisfying $f(\mu) = \mu$.

## 5 Short-term: Safety Guarantees

For the short-term running fairness property we require the shielded fairness process to stay within the running target $\mathcal{S} \subseteq [0, 1]$ at all finite times after the burn-in $\tau$ (see Ex. 2). To prove that our shield satisfies running fairness with high-probability we develop upper bounds on the probability and the expected number of point fairness violations over an interval.

**Definition 2.** Let $M$ be the shielded fairness process generated by the decision probability $p$ and an energy shield with energy function $\zeta$. For a running target interval $\mathcal{S} \subseteq [0, 1]$, we define two violation measures, the probability of violating point fairness $\mathcal{P}_\mathcal{S}$ and the expected number of point fairness violations $\mathcal{E}_\mathcal{S}$ over a time interval $[T, T'] \subseteq \mathbb{N} \cup \{\infty\}$, respectively, as

$$\mathcal{P}_\mathcal{S}(M_{T:T'}) = \mathbb{P}\left[\exists t \in [T, T'] : M_t \notin \mathcal{S}\right] \quad \text{and} \quad \mathcal{E}_\mathcal{S}(M_{T:T'}) = \sum_{t=T}^{T'} \mathbb{E}\left[\mathbf{1}[M_t \notin \mathcal{S}]\right].$$

### 5.1 Upper-bounds

Our main result is a bound for the probability of the shielded fairness process violating point fairness at time $T$. This result follows from constructing a martingale that upper-bounds the distance of the process to its converging point $\mu^*$, and using Azuma-Hoeffding's inequality on said martingale.

**Theorem 1.** Let $I = [L, U]$ such that $\kappa, p \in I$. Let $\tau = 4/\min(|L - \mu^*|, |U - \mu^*|)$, $K = (1/32) \cdot 4^\beta$, and $\beta = \sup_{r \in [0,1]} f'(r)$, then for every $t \geq \tau$ we have

$$\mathbb{P}(M_t \notin I) \leq \exp\left(-Kt|L - \mu^*|^2\right) + \exp\left(-Kt|U - \mu^*|^2\right). \tag{4}$$

Note that, by Eq. 2, $f$ is always non-increasing (see Fig. 1a), implying that $\beta \leq 0$ and $K \leq 1/32$.

**Property bounds.** Utilizing Theorem 1, the expected number of violations and the probability of a single violation in an interval, follows from Boole's inequality.

**Corollary 1.** Let $r_B = e^{-K|B - \mu^*|^2}$ for $B \in \{L, U\}$. For every $T, T' \in \mathbb{N}$ s.t. $\tau \leq T < T'$, then

$$\mathcal{E}_\mathcal{S}(M_{T:T'}) \leq \sum_{t=T}^{T'}(r_L^t + r_U^t), \qquad \text{and} \qquad \mathcal{E}_\mathcal{S}(M_{T:\infty}) \leq \frac{r_L^T}{1 - r_L} + \frac{r_U^T}{1 - r_U}.$$

Hence, it follows that $\mathcal{P}_\mathcal{S}(M_{T:T'}) \leq \mathcal{E}_\mathcal{S}(M_{T:T'})$ for $T' \in \mathbb{N} \cup \{\infty\}$ s.t. $T' \geq T$.

### 5.2 Monotonicity with Respect to the Energy Function

In this section, we define a partial order among energy functions given by their steepness, and show how our violation properties are monotone with respect to steepness.

**Definition 3.** Let $\zeta_1, \zeta_2$ be two energy functions. We say that $\zeta_1$ is *steeper* than $\zeta_2$, and denote it by $\zeta_1 \succeq \zeta_2$ if $\zeta_1(y) \geq \zeta_2(y)$ for all $y \in [0, 1]$.

Intuitively, a steeper energy function (see Fig. 1a) constitutes a more "aggressive" shield, leading to fewer safety violations, both in probability and in expectation.

**Theorem 2.** Let $\zeta_1 \succeq \zeta_2$ be two energy functions, with a common minimum at $\kappa$. Let $\mathcal{S} = [L, U]$. Let $M^{\zeta_1}$ and $M^{\zeta_2}$ be the shielded fairness process generated by enforcing the decision process of $p \in [0, 1]$ with $\zeta_1$ and $\zeta_2$, respectively. Let $\tau = \lceil \max\{1/|\kappa - L|, 1/|\kappa - U|\} \rceil$, for all $T \in \mathbb{N}$ and $T' \in \mathbb{N} \cup \{\infty\}$ such that $T < T'$ we have

$$\mathcal{E}_\mathcal{S}(M^{\zeta_1}_{T:T'}) \leq \mathcal{E}_\mathcal{S}(M^{\zeta_2}_{T:T'}), \qquad \text{and} \qquad \mathcal{P}_\mathcal{S}(M^{\zeta_1}_{T:T'}) \leq \mathcal{P}_\mathcal{S}(M^{\zeta_2}_{T:T'}).$$

# 6 LONG-TERM: LIVENESS GUARANTEES

For the long-term limit fairness property, we require the shielded fairness process to converge to a point in the limit target $\mathcal{L} \subseteq [0, 1]$. To prove that our shield satisfies limit fairness, we identify the conditions on the energy function to ensure that the shielded fairness process converges to a target value $\mu^*$ under the decision probability $p$ with an expected average interference cost of $|p - \mu^*|$.

**Main result.** It is remarkable that both the convergence of $M_t$ and the expected cost depend *only* on $p$ and the value of $\zeta$ at the target point $\mu^*$ and not on the pivot point $\kappa$.

**Theorem 3.** *Let $\mu^* \in [0, 1]$. Given the shielded decision process from Eq. 3, with bias $p$ and energy function $\zeta$, the shielded fairness process $(M_t)_{t \in \mathbb{N}}$ converges almost surely (a.s.) to $\mu^*$ if and only if*

$$\zeta(\mu^*) = \begin{cases} (\mu^* - p)/(1 - p) & \text{if } p < \mu^*, \\ (p - \mu^*)/p & \text{otherwise} \end{cases}. \tag{5}$$

*Furthermore, the expected interference cost $(\mathbb{E}[\nu_t])_{t \in \mathbb{N}}$ converges almost surely to $|\mu^* - p|$.*

**Proof intuition.** We establish that the shielded fairness process $M = (M_t)_{t \in \mathbb{N}}$ converges at the fixpoint of $f$. First, we show $f$ has indeed a unique fixed point located between $p$ and the pivot $\kappa$.

**Lemma 1.** *The function $f \colon [0, 1] \to [0, 1]$ defined as in Eq. 2 is continuously differentiable, and has a unique point $\mu^* \in [0, 1]$ such that $f(\mu^*) = \mu^*$. Furthermore, $\mu^*$ sits between $p$ and $\kappa$.*

Once we know $f$ has a unique fixed point, we use stochastic approximation theory to prove that the shielded fairness process $M$ converges almost surely to the fixed point.

**Lemma 2.** *The fairness process, defined in Eq. 3, converges a.s. to the unique fixpoint $\mu^*$ of $f$, as defined in Eq. 2. The error $(M_t - \mu^*)^2$ converges a.s. at the rate of $o(1/t^\lambda)$ for all $\lambda \in (0, 1)$.*

*Proof sketch.* Let $g(x) = f(x) - x$, then we rewrite the update rule in Eq. 3 as

$$M_{t+1} = M_t + \gamma_t \big( g(M_t) + \xi_{t+1} \big), \tag{6}$$

where $\xi_t = Z_t - f(M_{t-1})$, and $\gamma_t = 1/t$. This is a classical Robbins-Monro form for stochastic approximation Borkar (2008). From stochastic approximation theory we know that, under certain regularity conditions, $M_t$ from Eq. 6 approximates the zero value of $g$, which is fixed point of $f$. We use Karandikar & Vidyasagar (2024) to bound the convergence rate. $\square$

Equation 5 in Theorem 3 follows from Lemma 2 by noticing that the fixed point $\mu^*$ has to satisfy Eq. 7. Since the fixed point lies between $p$ and $\kappa$, the branch in Eq. 7 is chosen based on $\mu^* \leq p$.

$$\mu^* = \begin{cases} p + (1 - p) \cdot \zeta(\mu^*) & \text{if } \mu \leq \kappa, \\ p \cdot (1 - \zeta(\mu^*)) & \text{if } \mu > \kappa \end{cases}. \tag{7}$$

We show that the expected cost of intervention at step $t + 1$ is the probability of not seeing the favorable decision (1 when the current average is below $\kappa$ and 0 otherwise) and having an energy high enough to intervene. The expected intervention cost $h(\mu_t)$ converges $h(\mu_t) \to h(\mu^*)$, because the fairness value converges $\mu_t \to \mu^*$, where $h$ is defined as

$$h(\mu) = \begin{cases} (1 - p) \cdot \zeta(\mu) & \text{if } \mu \leq \kappa \\ p \cdot \zeta(\mu) & \text{otherwise} \end{cases}. \tag{8}$$

**Lemma 3.** *For the process described in Eq. 3 , the corresponding sequence of average interference $(\nu_t)_{t \in \mathbb{N}}$ converges to $h(\mu^*)$, where $\mu^*$ is the fixpoint of $f$ (Eq. 2) and $h$ is as defined in Eq. 8.*

Finally, Eq. 5 and Lemma 3 imply that the expected intervention cost converges to $|p - \mu^*|$.

# 7 SHIELD SYNTHESIS

In this section, we state the energy shield synthesis problem and propose Alg. 1 as a solution.

**Problem statement.** A problem instance $(\mathcal{V}, \Xi, p, \varphi, \delta, \varepsilon)$ consists of: a violation measure $\mathcal{V} \in \{\mathcal{P}, \mathcal{E}\}$, a totally ordered set of energy functions $\Xi = \{\zeta_k\}_{k \in R}$ indexed by some interval $R \subset \mathbb{R}$ where for all $i \leq j \in R$ we have $\zeta_i \preceq \zeta_j$, a decision probability $p \in [0, 1]$; a fairness target $\varphi = (\tau, \mathcal{S}, \mathcal{L})$, a violation threshold $\delta > 0$, and an approximation tolerance $\varepsilon > 0$. Given a problem instance, find the least invasive energy shield that satisfies the fairness violation constraint.

**Problem 1.** *Given $(\mathcal{V}, \Xi, p, \varphi, \delta, \varepsilon)$ find an energy function $\zeta \in \Xi$ such that:*
*(i) the shielded fairness process $M^\zeta$ with parameter $p$ satisfies*

$$\mathcal{V}_{\mathcal{S}}\zeta(M^\zeta_{\tau:\infty}) \leq \delta \quad and \quad \lim_{t \to \infty} M^\zeta_t \in \mathcal{L} \quad almost \; surely, \; and \tag{9}$$

*(ii) $\zeta$ is $\epsilon$-minimal, i.e., if $\zeta^\star \in \Xi$ is the smallest valid element, then $|\mathcal{V}_{\mathcal{S}}(M^{\zeta^\star}_{\tau:\infty}) - \mathcal{V}_{\mathcal{S}}(M^\zeta_{\tau:\infty})| \leq \epsilon$.*

We remark that if the violation measure is $\mathcal{P}$, then running fairness should be satisfied with probability greater than $1 - \delta$, and if it is $\mathcal{E}$, then the total number of point fairness violations after $\tau$ should be bounded by $\delta$. In Alg. 1 we show a synthesis procedure based on having a family of energy functions $\Xi = (\zeta_r)_{r \in R}$ that is indexed by some interval $R \subset \mathbb{R}$ and is monotonic with respect to the index. In Appendix B we describe such a family, denoted $(\zeta^{\text{Mon}}_{r;p,\mathcal{S},\mathcal{L}})_{r \in (0,1)}$, which is defined w.r.t. a decision probability $p$ and a specification $\varphi$, as a piecewise exponential and polynomial function.

---

**Algorithm 1** Shield synthesis

---

**Require:** problem instance $(\mathcal{V}, \Xi, p, \varphi, \delta, \varepsilon)$.
1: $l \leftarrow \min R; u \leftarrow \max R$          ▷ Set lower and upper bound for energy function w.r.t. $\prec$.
2: $T_{\text{DP}} \leftarrow \min\{t \in \mathbb{N} \mid \frac{r^t_-}{1-r_-} + \frac{r^t_+}{1-r_+} \leq \varepsilon\}$     ▷ smallest $t$ s.t. bound (Cor. 1) satisfies tolerance
3: **if** CONDITION$(\zeta_u, T_{\text{DP}}) > \delta$ **then return** FAIL ▷ The most strict energy function is not enough
4: **while** $l \neq u$ **do**
5:     $m \leftarrow (l+u)/2$ ;   $d \leftarrow$ CONDITION$(\zeta_m, T_{\text{DP}})$
6:     **if** $|d - \delta| < \epsilon$ **then return** $\zeta_m$
7:     **if** $d \leq \delta$ **then** $l \leftarrow m$ **else** $u \leftarrow m$

---

**Algorithm.** The algorithm exploits the monotonicity of the energy function family to find the least steep energy function that satisfies the violation condition, which determines the least invasive shield. Inside of the binary-search it is necessary to approximate the violation measure $\mathcal{V}$. The approximation algorithm, in CONDITION, uses dynamic programming and the tail-bounds from Section 5. Concretely, we divide the interval $[\tau, \infty)$ into two intervals: a prefix interval $[\tau, T_{\text{DP}}]$ and tail interval $(T_{\text{DP}}, \infty)$. For the prefix, we compute the exact violation measure with standard dynamic programming techniques, For the tail term, we use the bound in Sec. 5.1. Since the bounds for a violation in the tail can be made arbitrarily small for large enough $T_{\text{DP}}$, we use them to approximate the exact violation value with as much precision as required—a time-precision trade-off.

## 8 UNKNOWN AND NON-STATIONARY INPUT DISTRIBUTIONS

Until know we have assumed that the bias of the system $p$ is fixed an unknown. In this section, we explore what results are possible when relaxing said conditions.

**Setting 1: $p$ is fixed but unknown.** This corresponds to shielding a sequence sampled from a fixed distribution, with unknown bias $p$. In this case, we can use the natural estimator $\hat{p}_t = (1/t) \sum_i x_i$ as a replacement for $p$ and update our energy function at each step to reflect the current estimation of. Since $\hat{p}_t$ converges to $p$ a.s., the energy function we use also converges, and since it is design to make the shielded process converge to a certain target, we retain the long-term a.s. guarantees.

**Theorem 4.** *Let $\mu^* \in [0, 1]$ be a fairness target. Let $(\zeta_q)_{q \in [0,1]}$ be a family of energy functions satisfying*

$$\zeta_q(\mu^*) = \begin{cases} (\mu^* - q)/(1 - q) & if \; q < \mu^*, \\ (q - \mu^*)/q & otherwise. \end{cases} \tag{10}$$

*Then the shielded process that takes at each step the energy function $\zeta_{\hat{p}_t}$ converges a.s. to $\mu^*$.*

**Setting 2:** $(p_t)_{t \in \mathbb{N}}$ **is varying and unknown.** If $(p_t)_{t \in \mathbb{N}}$ is allowed to evolve arbitrarily, a non-trivial shield cannot guarantee convergence for the shielded process to a target point. The best effort solution we can give is that a fixed energy function, if it is steep enough, can guarantee that the process stays within a certain target interval almost surely.

**Theorem 5.** *Let* $\zeta \colon [0,1] \to [0,1]$ *be an energy function, and* $L, R$ *the unique point satisfying*

$$L < \kappa \ \text{and} \ \zeta(L) = L, \qquad R > \kappa \ \text{and} \ \zeta(R) = 1 - R.$$

*Then, for the shielded process* $(M_t)_{t \in \mathbb{N}}$ *it holds almost surely that*

$$\liminf_{t \to \infty} M_t \geq L, \qquad \text{and} \qquad \limsup_{t \to \infty} M_t \leq R.$$

# 9 ENERGY SHIELDS FOR GROUP FAIRNESS

Group fairness formalizes equal treatment of demographic groups in binary decision settings Mehrabi et al. (2021). Group fairness metrics typically compare decision probabilities across groups. For example, demographic parity ensures that the ratio of positive decisions is independent of the group. With a tolerance of $\varepsilon > 0$, this means

$$\mathbb{P}(X = 1 \mid \text{Group} = A) - \mathbb{P}(X = 1 \mid \text{Group} = B) \in [-\varepsilon, +\varepsilon]. \tag{11}$$

Probabilities are taken w.r.t. the population distribution and a decision maker.

In situations where the inputs come in pairs and we are forced to give a positive decision to one group and a negative decision to the other group, the fairness-relevant part of the sequence can be modeled with the stochastic processes described in Sec. 3. This is illustrated in Ex. 1.

**Two-group setting.** A more common sequential input setting is that at each time an instance of either group $A$ or group $B$ is presented, and the decision can be either positive or negative for that instance. To model group fairness in such problems, we need to extend our input space $X$ to be $\mathbb{G} \times \mathbb{B}$, where $\mathbb{G} = \{A, B\}$. At each point in time, we obtain an input $(g_t, x_t)$, where $g_t \in \{A, B\}$ indicates whether the input belongs to the demographic group $A$ or $B$, and $x_t \in \{0, 1\}$ indicates whether the initial decision is positive or not. The value of $g_t$ is sampled from a distribution Bernoulli($\pi$), where $\pi$ indicates the probability of seeing an instance of group $A$. The probability of a positive decision $x_t$ is group dependent, i.e., $x_t$ is sampled from a distribution Bernoulli($p_{g_t}$), where $p_A$ and $p_B$ indicate the positive decision probability for each group, respectively. The values of $\pi \in (0, 1)$, and $p_A, p_B \in [0, 1]$ are the parameters of the setting.

The shield can modify the decision, but not the demographic group. Therefore a (probabilistic) shield is now a function $\mathfrak{S} \colon (\mathbb{G} \times \mathbb{B} \times \mathbb{B})^* \times (\mathbb{G} \times \mathbb{B}) \to [0, 1]$ mapping a history $(g_i, x_i, y_i)_{i=1}^{t-1}$ and an input $(g_t, x_t) \in \mathbb{G} \times \mathbb{B}$ to a nudging probability $q_t$, defining the distribution from which the next intervention is samples, i.e., $Y_t = \text{Bernoulli}(q_t)$.

**Updated problem sketch.** We consider the group process $G = (G_t)_{t \in \mathbb{N}}$, the decision process $X = (X_t)_{t \in \mathbb{N}}$ the intervention process $Y = (Y_t)_{t \in \mathbb{N}}$, the shielded input process $Z = (Z_t)_{t \in \mathbb{N}}$, and the shielded fairness process $M = (M_t)_{t \in \mathbb{N}}$, defined as follows:

$$G_t \sim \text{Bernoulli}(\pi), \quad X_t \sim \text{Bernoulli}(p_{G_t}), Y_t = \text{Bernoulli}(\mathfrak{S}(G_1, X_1, Y_1, \dots, Y_{t-1}, G_t, X_t))$$

$$Z_t = (1 - X_t) \cdot Y_1 + X_t \cdot (1 - Y_t), \quad M_t = \frac{\sum_{i=1}^{t} Z_i \cdot \mathbf{1}[G_i = A]}{\sum_{i=1}^{t} \mathbf{1}[G_i = A]} - \frac{\sum_{i=1}^{t} Z_i \cdot \mathbf{1}[G_i = B]}{\sum_{i=1}^{t} \mathbf{1}[G_i = B]}$$

Given a fairness target $\varphi = (\tau, \mathcal{S}, \mathcal{L})$ and a probability $\delta \in [0, 1]$ the goal of the shield is to guarantee

$$\mathbb{P}(\forall t \geq \tau \colon M_t \in \mathcal{S}) \geq 1 - \delta, \quad \text{and} \quad \mathbb{P}(\lim_{t \to \infty} M_t \in \mathcal{L}) \geq 1 - \delta.$$

Note that now the fairness measure $M_t$ can take values in the interval $[-1, +1]$, so we can expect the fairness targets to reflect that, i.e., in general $\mathcal{S}, \mathcal{L} \subseteq [-1, +1]$.

**Implementation of the energy shield.** In this setting, an energy function follows the same definition as Def. 1, with the only modification that the domain changes from $[0, 1]$ to $[-1, +1]$, so $\zeta \colon [-1, +1] \to [0, 1]$.

The shield monitors the evolution of the shielded fairness process $M$. At each time $t$, the current fairness value is $\mu_t$. If $\mu_t > \kappa$, the shield favours the decision that tends to decrease $\mu_t$, and if $\mu_t < \kappa$, the shield favours the decision that tends to increase $\mu_t$. When the input is $(g_t = A, x_t, y_t)$, forcing $z_t = 1$ increases the value of $\mu_t$, and forcing $z_t = 0$ decreases it. On the other hand, when $g_t = B$, forcing $z_t = 1$ decreases the value of $\mu_t$, and forcing $z_t = 0$ increases it.

The shield is implemented with the same rationale as in the setting with a single process: if the proposed decision agrees with the direction favoured by the shield, the shield accepts it. Otherwise, the shield may flip the decision with a probability given by $\zeta(\mu_t)$.

In this setting, we obtain similar long-term guarantees as with Thm. 3.

**Theorem 6.** *Let $\mu^* \in [-1, +1]$. The two-group shielded fairness process $(M_t)_{t \in \mathbb{N}}$ converges a.s. to $\mu^*$ if and only if*

$$\zeta(\mu^*) = \begin{cases} (\mu^* - (p_A - p_B))/(1 - (p_A - p_B)) & \text{if } p_A - p_B < \mu^*, \\ (p_A - p_B - \mu^*)/(p_A - p_B) & \text{otherwise.} \end{cases} \tag{12}$$

*The expected interference cost $(\mathbb{E}[\nu_t])_{t \in \mathbb{N}}$ converges a.s. to a value smaller than $|\mu^* - (p_A - p_B)|$.*

## 10 EXPERIMENTS

We evaluate the impact of energy function on the fairness measure in the setting of Ex. 2, explore the time-precision trade-off in Alg. 1, and compare against existing fairness shields.

**Energy functions.** We consider a fair ($p = 0.5$) and biased ($p = 0.65$) decision maker with fairness target $\varphi = (\tau, [0.4, 0.6], \{0.5\})$. In addition to considering polynomial $\zeta^{\text{Pol}}$ and exponential $\zeta^{\text{Exp}}$ energy functions (see Ex. 4), we consider $\zeta^{\text{Idle}}$, which is always 0, and $\zeta^{\text{Naive}}$, which is 0 in the interior of the running target and 1 elsewhere. In Fig. 1a we can observe that: if the fixpoint of the functions–indicated by the intersections of $45°$-line with the value graph of $f$–is contained within the running target, the number of violations decays quickly; if the decision maker is biased, a pivot within a given target interval does not guarantee the fairness value convergence to a point in the target. if the decision maker is fair, our shields remain useful by further reducing violations.

**Approximation precision.** The synthesis Algorithm 1 performs a violation probability approximation (VPA) by running dynamic programming (DP) up to a horizon $T_{\text{DP}}$ and then utilizing the infinite horizon tail-bound from Sec. 5. Fig. 1b investigates the precision-time trade-off. We observe that an extension of the DP horizon $T_{\text{DP}}$ leads to an exponential gain in VPA precision, at the cost of a quadratic increase in computation time. This impacts the runtime of Alg. 1, as higher precision requirements demands longer DP horizons. The culprit is the looseness of the tail-bounds over the short horizon, as demonstrated by the gain in precision, if larger burn-ins are considered.

**Comparison.** We benchmark our energy shields w.r.t. $\zeta^{\text{Mon}}_{0.1; p, \mathcal{S}, \mathcal{L}}$ against two baselines: the naive shield (Ex. 3), which enforces running fairness strictly, and periodic shields Cano et al. (2025b), which requires heavy-computation at runtime to enforce point fairness periodically with optimal expected cost. For a fairness target $\varphi = (100, [0.4, 0.6], [0.49, 0.51])$ and decision probability $p = 0.3$, our shield (tuned to $\mu^* = 0.5$) achieves both running and limit fairness, unlike baselines, which do not support long-term guarantees. When configured for running fairness, baselines cluster at target boundaries without converging; when tuned for limit fairness, periodic shields over-intervene and naive shield are almost maximally invasive. This is reflected in the intervention costs: baselines prioritizing running fairness intervene less, while those targeting limit fairness intervene more.

## 11 DISCUSSION

**Safety and liveness.** Formal verification classifies properties over infinite traces as safety or liveness Lamport (1977); Henzinger et al. (2023b). A safety property asserts that "bad" things never

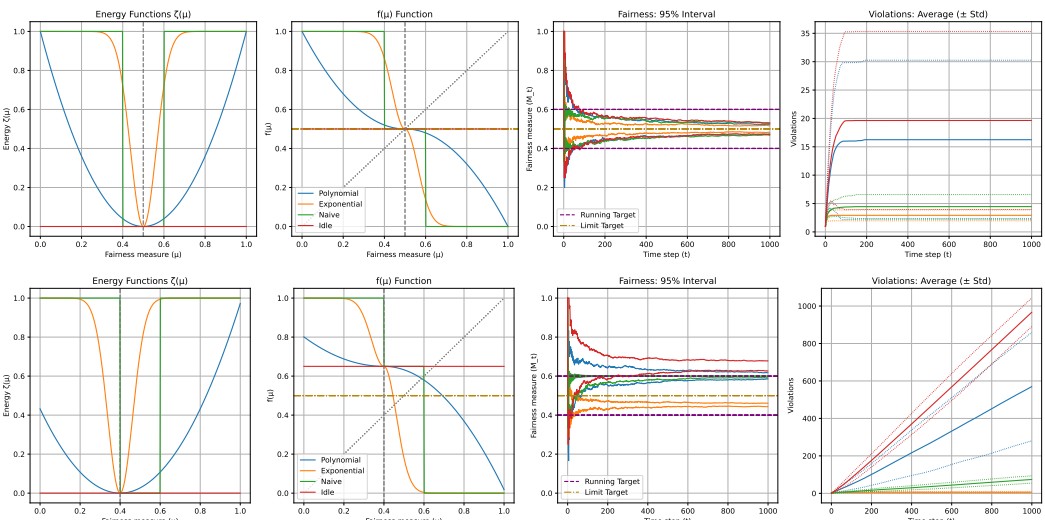

(a) Simulated impact of the energy functions $\zeta$ for fairness target $\varphi = (0, [0.4, 0.6], \{0.5\})$, decision probability $p$. **Rows (R):** (R1) $p = 0.5$ and $\zeta \in \{\zeta^{\text{Naive}}, \zeta^{\text{Idle}}, \zeta^{\text{Pol}}_{0.5,4,2}, \zeta^{\text{Exp}}_{0.5,1,128}\}$; (R2) considers $p = 0.65$ and $\zeta \in \{\zeta^{\text{Naive}}, \zeta^{\text{Idle}}, \zeta^{\text{Pol}}_{0.4,2.7,2}, \zeta^{\text{Exp}}_{0.4,1,128}\}$. **Columns (C):** (C1) the behavior of the energy function for each fairness value. (C2) illustration of the characteristic functions $f$ and their respective fixpoints $f(\mu) = \mu$. (C3) 95% confidence interval of fairness values at each time. (C4) Average point fairness violations with standard deviation at each time. The simulation results are averaged over 1000 simulations for each $p$ and $\zeta$.

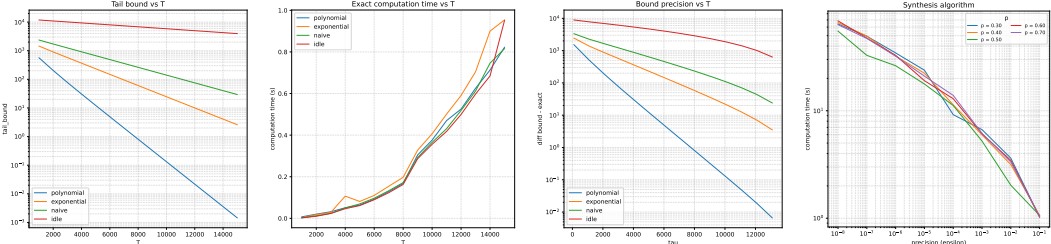

(b) Time-precision trade-off in the violation probability approximation (VPA) of Alg. 1 with fairness target $\varphi = (100, [0.3, 0.7], \{0.45, 0.55\})$, decision probability $p = 0.65$, and energy function $\zeta \in \{\zeta^{\text{Naive}}, \zeta^{\text{Idle}}, \zeta^{\text{Pol}}_{0.4,2.7,2}, \zeta^{\text{Exp}}_{0.4,1,128}\}$. **Columns (C):** (C1) VPA with increasing until dynamic programming (DP) threshold $T_{\text{DP}}$. (C2) Computation time as $T_{\text{DP}}$ increases. (C3) VPA precision, i.e., VPA with $T_{\text{DP}} = 0$ compared to $T_{\text{DP}} = 15000$, for increasing burn-ins $\tau$. (C4) Runtime time of Alg. 1 as precision $\varepsilon$ increases, for different decision probabilities $p$, with fixed $\delta = 0.1$.

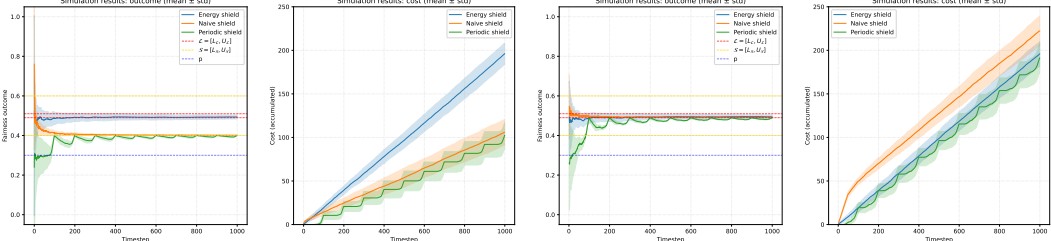

(c) Energy shields compared against naive and periodic shield, for fairness target $\varphi = (\tau = 100, \mathcal{S} = [0.4, 0.6], \mathcal{L} = [0.49, 0.51])$ and decision probability $p = 0.3$. As energy function, we use $\zeta^{\text{Mon}}_{0.1;p,\mathcal{S},\mathcal{L}}$. **Columns (C):** The naive and periodic shield are tuned for the: (C1&2) running target $[0.4, 0.6]$; (C3&4) limit target $[0.49, 0.51]$. (C1&C3) depicts the fairness measure and (C2&C4) the accumulated number of interventions.

Figure 1: Experimental evaluation of energy shields: Fig. 1a shows the impact of different energy functions; Fig. 1b shows time-precision trade-offf in Alg. 1; Fig. 1c shows the comparison with existing fairness shields.

occur, e.g., the car never crashes, thus its violation can be observed on a finite prefix. A liveness property asserts that "good" things will eventually happen, e.g., the car reaches its goal, thus its satisfaction can only be determined in the limit, and every finite trace can be extended to an infinite satisfiable trace. For stochastic processes one evaluates the satisfaction probability of the safety or liveness property w.r.t. the law of the process Baier & Katoen (2008). We deliberately connected short- and long-term fairness to safety and liveness respectively. The short-term running fairness property requires the fairness measure to remain within the running target at all times, which implies that once violated it remains violated along the infinite sequence, i.e., safety. The long-term limit fairness property requires the limit of the fairness measure to lie in the limit target, which implies it can only be determined in the limit, i.e., liveness. The convergence in the limit holds because the fairness measure is an average. This normalization allows the distance to $\mu^*$ to shrink and the process to converge almost surely. We emphasize that this implies that almost surely the fairness measure eventually remains within the limit target. It does not imply that the fairness measure eventually remains within the limit target almost surely. Formally, $\mathbb{P}(\exists \tau \in \mathbb{N} \forall t \geq \tau \colon M_t \in \mathcal{L}) = 1$ is satisfied, and $\exists \tau \in \mathbb{N} \colon \mathbb{P}(\forall t \geq \tau \colon M_t \in \mathcal{L}) = 1$ is not.

## 12 CONCLUSION

We introduced *energy shields*, a physics-inspired framework for enforcing fairness at runtime. Energy shields are a lightweight, probabilistic mechanisms that provide rigorous short- and long-term fairness guarantees. Utilizing a bowl-shaped energy function, enforcing fairness is reduced to an energy minimization problem, which enables gentle, adaptive interventions. We provide a synthesis algorithm based on binary search and dynamic programming to find the least-invasive shield for a given specification. The experimental validation demonstrates the practicality of our synthesis procedure and supports, in the comparison with Cano et al. (2025b), the claim that energy shields are the first shields to satisfy both short- and long-term guarantees. While our results establish a foundational theory of energy-based fairness shielding, extending these guarantees to multi-group fairness, properties beyond fairness and dynamic environments remains an exciting direction.

## DISCLAIMERS

### ETHICS STATEMENT

This work is strongly motivated by the problem of ensuring algorithmic fairness in machine learning applications. Although this is a sensitive topic, the ethical considerations for this paper are minimal. Our contributions are largely theoretical, and we do not employ any datasets containing sensitive information.

### REPRODUCIBILITY STATEMENT

The proofs for all theoretical results are stated in the paper. For most results, the main body of the text contains only a sketch of the argument and the full proof can be found in the appendix. The code required to reproduce the experiments, including the synthesis procedure, are provided in the supplementary materials. Our empirical evaluation requires only models computational resources. All computation time experiments reported were performed with a Macbook with the M2 chip.

### USE OF LLMs

LLM tools were used to polish the text, to brainstorm arguments for proving the theoretical results, and to generate parts of the code used for the experimental evaluation.

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

## A  DETAILED PROOFS

**Claim 1** (Shielded decision process). *The shielded decision process generated by the energy shield can be written as a sequence of Bernoulli random variables with evolving bias, $Z_t \sim \text{Bernoulli}(p_t)$. The biases are defined recursively as $p_1 = 1$ and $p_{t+1} = f(\mu_t)$ for a given history $z_1, \ldots, z_t$, where*

$$f(\mu) = \begin{cases} p + (1-p)\zeta(\mu) & \text{if } \mu \le \kappa, \\ p \cdot (1 - \zeta(\mu)) & \text{if } \mu > \kappa. \end{cases} \tag{13}$$

*Moreover, the resulting shielded fairness process can be written as*

$$M_t = M_{t-1} + \frac{1}{t}(Z_t - M_{t-1}) \quad \text{(with } M_0 = 0\text{)}. \tag{14}$$

*Proof.* Both equations are just simple computations.

**Equation 13.** Suppose $\mu_t \le \kappa$, i.e., the shield favors 1's. Then a 1 can be obtained either by $X_{t+1} = 1$ (which happens with probability $p$), or by flipping the decision $X_{t+1} = 0$ (which happens with probability $(1-p)\zeta(\mu_t)$). Therefore, when $\mu_t \le \kappa$, $Z_{t+1}$ behaves like a Bernoulli of bias $p + (1-p)\zeta(\mu_t)$. Analogously, when $\mu_t > \kappa$, the shield favors 0's, so to obtain a 1 we need to toss $X_{t+1} = 0$ and for the shield to fail to flip the decision, which happens with probability $(1 - \zeta(\mu_t))$.

**Equation 14.**

$$M_t = \frac{\sum_{i=1}^t Z_i}{t} = \frac{Z_t}{t} + \frac{t-1}{t} \cdot \frac{\sum_{i=1}^{t-1} Z_i}{t-1} = \frac{Z_t}{t} + \frac{t-1}{t} M_{t-1} = M_{t-1} + \frac{1}{t}(Z_t - M_{t-1}).$$

$\square$

### A.1  LONG TERM GUARANTEES

**Lemma 1.** *The function $f : [0,1] \to [0,1]$ defined as in Eq. 2 is continuously differentiable, and has a unique point $\mu^* \in [0,1]$ such that $f(\mu^*) = \mu^*$. Furthermore, $\mu^*$ sits between $p$ and $\kappa$.*

*Proof.* We need to prove smoothness, existence of the fixpoint, and that it sits between $p$ and $\kappa$.

*Smoothness.* The function $f$ inherits continuous differentiability from $\zeta$ clearly at all points except maybe $\mu = \kappa$. For $\mu = \kappa$, the assumption of the energy function being flat at the pivoting point $\kappa$ guarantees continuous differentiability (Def. 1, item 3). We first write the expression for $f'$:

$$f'(\mu) = \begin{cases} -p \cdot \zeta'(\mu) & \text{if } \mu > \kappa \\ (1-p)\zeta'(\mu) & \text{otherwise.} \end{cases} \tag{15}$$

We can check that $f$ is continuous at $\mu = \kappa$, as

$$p \cdot (1 - \zeta(\kappa)) = p, \text{ and } p + (1 - p) \cdot \zeta(\kappa) = p.$$

Similarly, we can check that $f'$ is continuous at $f = \kappa$, as

$$-p \cdot \zeta'(\kappa) = 0, \text{ and } (1 - p) \cdot \zeta'(\kappa) = 0.$$

Note that both are necessary conditions. For $f$ to be continuous at $\mu = \kappa$, we need

$$p(1 - \zeta(\kappa)) = p + (1 - p)\zeta(\kappa) \iff -p\zeta(\kappa) = \zeta(\kappa) - p\zeta(\kappa) \iff \zeta(\kappa) = 0.$$

Similarly, for $f'$ to be continuous as $\mu = \kappa$, we need

$$-p\zeta'(\kappa) = (1 - p)\zeta'(\kappa) \iff \zeta'(\kappa) = 0.$$

*Existence and uniqueness of a fixpoint.* Consider the function $g(\mu) = f(\mu) - \mu$. We have $g(0) = p(1 - \zeta(0)) - 0 > 0$ and $g(1) = p + (1 - p)\zeta(1) - 1 < 1 - 1 = 0$. Since $g$ is continuous, there must be a point $\mu^* \in [0, 1]$ such that $g(\mu^*) = 0$. Suppose this point is not unique, i.e., there exist two point $x < \mu$ such that $g(x) = g(\mu) = 0$. As we can see in Eq. equation 15, $f'(\mu) \le 0$ for all $\mu \in [0, 1]$, therefore $g$ is non-increasing, so if $g(x) = g(\mu) = 0$, then $g(z) = 0$ for all $z \in [x, \mu]$. The interval $[x, \mu]$ contains at least two numbers that are either larger or lower than $\kappa$. Let $z_1, z_2 \in [x, \mu] \cap [0, \kappa)$ be such numbers, with $z_1 < z_2$. Then we have

$$p + (1 - p)\zeta(z_1) - z_1 = p + (1 - p)\zeta(z_2) - z_2,$$

which implies that

$$z_2 - z_1 = (1 - p)(\zeta(z_2) - \zeta(z_1)).$$

On the left-hand side we have a positive number (because $z_1 < z_2$). On the right-hand side we have a non-positive number (because $\zeta$ is decreasing in the range $[0, \kappa)$, which is a contradiction. Since assuming $z_1, z_2 \in [x, \mu] \cap [0, \kappa)$ leads to a contradiction, it must be that $z_1, z_2 \in [x, \mu] \cap (\kappa, 1]$. However, with a similar argument this implies that

$$p(1 - \zeta(z_1)) - z_1 = p(1 - \zeta(z_2)) - z_2 \iff z_2 - z_1 = p(\zeta(z_1) - \zeta(\zeta_2)).$$

Since $\zeta$ is in the decreasing range, in the right-hand side we have again a non-positive term, which is a contradiction.

*Placement of the fixpoint.* We write the case where $p \le \kappa$, the case where $p > \kappa$ is analogous. If $p \le \kappa$, we have

$$g(p) = (1 - p)\zeta(p) \ge 0, \quad \text{and} \quad g(\kappa) = p - \kappa \le 0.$$

By continuity, there exists a point $\mu^* \in [p, \kappa]$ such that $g(\mu^*) = \mu^*$. Note that this includes the special case where $p = \kappa$, in which the unique fixpoint is $\mu^* = \kappa = p$. □

**Lemma 2.** *The process described in Eq. 3 converges almost surely to the unique fixpoint $\mu^*$ of $f$, as defined in Eq. 2. Furthermore, the convergence rate of the error satisfies $(M_t - \mu^*)^2 = o(1/t^\lambda)$ for all $\lambda \in (0, 1)$ almost surely.*

*Proof. Error construction.* For convenience, we will prove that the sequence converges to the unique root of $g(\mu) = f(\mu) - \mu$. Let $(V_t)_{t \in \mathbb{N}}$ be the sequence of squared distances to the target, i.e., $V_t = (\mu_t - \mu^*)^2$. We will prove that $V_t \to 0$ a.s. (almost surely).

First we need to find a recurrence formula for $V_t$. Recall from Eq. 6 the recurrence for $\mu_t$ is

$$\mu_t = \mu_{t-1} + \gamma_t(g(\mu_{t-1}) + \xi_t),$$

where $\gamma_t = 1/t$ and $\xi_t$ has null expectation. For $V_t$ we have

$$V_t = (\mu_t - \mu^*)^2 = (\mu_{t-1} - \mu^* + \gamma_t(g(\mu_{t-1}) + \xi_t))^2 \tag{16}$$

$$= (\mu_{t-1} - \mu^*)^2 + 2\gamma_t(\mu_{t-1} - \mu^*)(g(\mu_{t-1}) + \xi_t) + \gamma_t^2(g(\mu_{t-1}) + \xi_t)^2. \tag{17}$$

Taking conditional expectations we have

$$\mathbb{E}[V_t \mid \mu_{t-1}] = V_{t-1} + 2\gamma_t(\mu_{t-1} - \mu^*) + \gamma_t^2\mathbb{E}[(g(\mu_{t-1}) + \xi_t)^2]. \tag{18}$$

Since $g(\mu_{t-1})$ is determinstic given $\mu_{t-1}$ and $\mathbb{E}[\xi_t \mid \mu_{t-1}] = 0$, we have

$$\mathbb{E}[(g(\mu_{t-1}) + \xi_t)^2] = g(\mu_{t-1})^2 + \sigma_t^2,$$

where we define $\sigma_t^2 = \mathbb{E}[\xi_t^2 \mid \mu_{t-1}]$. Recall that $\xi_t = Z_t - f(\mu_{t-1})$, which is always in the interval $[-1, 1]$, therefore $\sigma_t^2 \leq 1$. Plugging everything into Eq. equation 17, we have

$$\mathbb{E}[V_t \mid \mu_{t-1}] \leq V_{t-1} + 2\gamma_t(\mu_{t-1} - \mu^*)g(\mu_{t-1}) + \gamma_t^2(g(\mu_{t-1})^2 + \sigma_t^2). \tag{19}$$

Let $\alpha_t = -2\gamma_t \frac{g(\mu_{t-1})}{\mu_{t-1} - \mu^*}$ and $\beta_t = \gamma_t^2(g(\mu_{t-1})^2 + \sigma_t^2)$. Note that $\alpha_t$ is well defined, even when $\mu_{t-1} = \mu^*$ as, using L'Hôpital's rule:

$$\lim_{\mu \to \mu^*} \frac{g(\mu)}{\mu - \mu^*} = \lim_{\mu \to \mu^*} \frac{f(\mu) - \mu}{\mu - \mu^*} = \lim_{\mu \to \mu^*} \frac{g'(\mu)}{1} = g'(\mu^*).$$

Also, $g$ is decreasing around $\mu^*$, so $g(\mu_{t-1})$ and $\mu_{t-1} - \mu^*$ have opposite signs. Therfore $\alpha_t \geq 0$. We can then rewrite Eq. equation 19 as

$$\mathbb{E}[V_t \mid \mu_{t-1}] \leq (1 - \alpha_t)V_{t-1} + \beta_t. \tag{20}$$

*Proof of convergence.* This is a standard form for the condition in the classical Robbins-Siegmund theorem Robbins & Siegmund (1971), which guarantees convergence of $V_t$ almost surely whenever the sequences $(\alpha_t)$ and $(\beta_t)$ are non-negative and satisfy the limiting properties that $\sum_{t=1}^{\infty} \alpha_t = \infty$ and $\sum_{t=1}^{\infty} \beta_t < \infty$ almost surely. We use the recent results in Karandikar & Vidyasagar (2024) to guarantee convergence. In particular, we use Thm. 5.1 to guarantee $V_t \to \infty$ almost surely, and Thm. 5.2 to guarantee rates of convergence. To apply both theorems, we need to guarantee that

1. $\sum_{t=1}^{\infty} \alpha_t = \infty$ almost surely, and

2. $\sum_{t=1}^{\infty} < \infty$ almost surely.

Note that both results are true in the strict sense (not only "almost surely"). In the case of $\alpha_t$, note that there exists $C > 0$ such that $g(\mu_t)/(\mu^* - \mu_t) \geq C$. Otherwise, it would imply that $g$ has a zero other than $\mu^*$ or that $g'(\mu^*) = 0$, both which we know are not true. Therefore

$$\sum_{t=1}^{\infty} \alpha_t \geq 2C \sum_{t=1}^{\infty} \gamma_t = \infty.$$

Since $\gamma_t = 1/t$, the sum $\sum_t \gamma_t$ diverges. For the case of $\beta_t$, recall by definition that $g(\mu_t)^2 \leq 1$. We have also previously established that $\sigma_t^2 \leq 1$:

$$\sum_{t=1}^{\infty} \beta_t \leq 2 \sum_{t=1}^{\infty} \gamma_t^2 < \infty.$$

Theorem 5.1 in Karandikar & Vidyasagar (2024) is stated in terms of a sequence $h_t$. We can use $h_t = V_t$ (since the identity is what Karandikar & Vidyasagar (2024) defines as a class $\mathcal{B}$ function), to guarantee $V_t \to 0$ almost surely.

*Convergence rates.* From Theorem 5.2 in Karandikar & Vidyasagar (2024), we have that $V_t = o(1/t^\lambda)$ for any $\lambda$ such that $\alpha_t \geq \lambda/t$ for sufficiently large $t$. In our case, we can take $\lambda = 2|g'(\mu^*)| - \epsilon$ for any $\epsilon > 0$. Therefore, $V_t = o(1/t^\lambda)$ for any $\lambda \in (0, 1) \cap (0, 2|g'(\mu^*)|) = (0, \min\{1, 2|g'(\mu^*)|\})$.

However, we can be more fine grained and show that actually $\min\{1, 2|g(\mu^*)|\} = 1$. Recall from the definition of $g$ that $g'(\mu) = f'(\mu) - 1$, and $f'(\mu) \leq 0$ for all $\mu \in [0, 1]$. Therefore $|g'(\mu)| = 1 + |f'(\mu)| \geq 1$. Altogether, we obtain the expected result of $V_t = o(1/t^\lambda)$ for all $\lambda \in (0, 1)$. $\square$

**Lemma 3.** *For the process described in Eq. 3 , the corresponding sequence of average interference $(\nu_t)_{t \in \mathbb{N}}$ converges to $h(\mu^*)$, where $\mu^*$ is the fixpoint of $f$ (Eq. 2) and $h$ is as defined in Eq. 8.*

*Proof.* By definition $N_t = \frac{1}{t} \sum_{i=1}^{t} Y_i$.

Since $\mathbb{E}[Y_t \mid \mu_{t-1}] = h(\mu_{t-1})$, and $\mu_t \xrightarrow{t \to \infty} \mu^*$ almost surely. Since $h$ is a continuous function,

$$\lim_{t \to \infty} N_t = h \left( \lim_{t \to \infty} M_t \right) = h(\mu^*).$$

$\square$

**Theorem 3.** *Let $\mu^* \in [0, 1]$. Given the shielded decision process as described in Eq. 3, with bias parameter $p$ and energy function $\zeta$, the shielded fairness process $(M_t)_{t \in \mathbb{N}}$ converges almost surely to $\mu^*$ if and only if*

$$\zeta(\mu^*) = \begin{cases} |\mu^* - p|/(1-p) & \text{if } p < \mu^*, \\ |\mu^* - p|/p & \text{otherwise.} \end{cases} \tag{21}$$

*Furthermore, the expected interference cost $(\mathbb{E}[\nu_t])_{t \in \mathbb{N}}$ converges almost surely to $|\mu^* - p|$.*

*Proof.* We know from Lemma 2 that the fairness outcome converges to a fixpoint that is always between $p$ and $\kappa$. If we want the process to converge to a desired outcome $\mu^*$, then we need to find $\kappa$ and $\zeta$ such that $f(\mu^*) = \mu^*$.

The function $f(\mu)$ is defined by parts depending on whether $\mu$ is larger or smaller than $\kappa$. If $p \leq \mu^*$, then we need $\kappa \geq \mu^*$, so we will be using the expression $f(\mu) = p + (1-p)\zeta(\mu)$ for $\mu = \mu^*$. Imposing $\mu^*$ to be a fixpoint, we have

$$\mu^* = p + (1-p)\zeta(\mu^*) \iff \zeta(\mu^*) = \frac{\mu^* - p}{1 - p}.$$

Analogously, if $p \geq \kappa$, then we need $\kappa \leq \mu^*$, so we will use the expression $f(\mu) = p(1 - \zeta(\mu))$. If we set $\mu^*$ to be a fixpoint, we have

$$\mu^* = p(1 - \zeta(\mu^*)) \iff \zeta(\mu^*) = 1 - \frac{\mu^*}{p}.$$

To prove the expectation of cost, just recall from Lemma 3 that the expected cost converges to $h(\mu^*)$. If $p \leq \mu^*$, then $\kappa \geq \mu^*$, so we have

$$h(\mu^*) = (1-p)\zeta(\mu^*), \quad \text{and} \quad \zeta(\mu^*) = \frac{\mu^* - p}{1 - p} \implies h(\mu^*) = \mu^* - p.$$

Analogously, if $p \geq \mu^*$, then $\kappa \leq \mu^*$, so we have

$$h(\mu^*) = p\zeta(\mu^*), \quad \text{and} \quad \zeta(\mu^*) = 1 - \frac{\mu^*}{p} \implies h(\mu^*) = p - \mu^*.$$

This completes the proof, since in both cases $h(\mu^*) = |\mu^* - p|$. $\square$

### A.2 SHORT-TERM GUARANTEES: UPPER BOUNDS

**Lemma 4.** *Let $\Delta_t = \mu_t - \mu^*$. Recall $\xi_t = Z_t - f(M_{t-1})$ from Equation 6 The following holds:*

$$\Delta_t = A_{1,t} \Delta_1 + \sum_{i=1}^{t-1} w_{i+1,t} \, \xi_{i+1}, \qquad \text{where} \tag{22}$$

$$A_{i,t} = \prod_{j=i}^{t-1} \left( 1 - \frac{\alpha_j}{j+1} \right), \qquad w_{i,t} = \frac{A_{i,t}}{i}, \qquad \text{and} \qquad \alpha_t := 1 - \frac{f(\mu_t) - \mu^*}{\mu_t - \mu^*}. \tag{23}$$

*Furthermore, the following inequality holds:*

$$\sum_{i=1}^{t-1} w_{i+1,t}^2 \leq \frac{4^{1-\beta}}{t+1} \qquad \text{with} \qquad \beta = \sup_{r \in [0,1]} f'(r). \tag{24}$$

*Proof.* From the definition of $\Delta_t$ and the recursion for $\mu_t$ expressed in Equation 3, we have

$$\Delta_{t+1} = \left(1 - \frac{\alpha_t}{t+1}\right)\Delta_t + \frac{\xi_{t+1}}{t+1}. \tag{25}$$

Equation 22 follows by induction on $t$ from Eq. 25. The base case $t = 1$ is trivial, since product defining $A_{i,t}$ is empty, and so is the sum of $w_{i+1,t}\xi_{i+1}$. For the induction step, can apply the induction hypothesis to Equation 25 to obtain

$$\Delta_{t+1} = \left(1 - \frac{\alpha_t}{t+1}\right)\Delta_t + \frac{\xi_{t+1}}{t+1}$$

$$= \left(1 - \frac{\alpha_t}{t+1}\right)\left(A_{1,t}\,\Delta_1 + \sum_{i=1}^{t-1} w_{i+1,t}\,\xi_{i+1}\right) + \frac{\xi_{t+1}}{t+1}.A_{1,t}\left(1 - \frac{\alpha_t}{t+1}\right)\Delta_1 +$$

Note that, from the definition of $A_{i,t}$, we have

$$\left(1 - \frac{\alpha_t}{t+1}\right)A_{i,t} = A_{i,t+1}, \qquad \text{and} \qquad \left(1 - \frac{\alpha_t}{t+1}\right)w_{i,t} = w_{i,t+1}.$$

Using the previous identities, plus the case that $w_{t+1,t+1} = \frac{1}{t+1}$ we get

$$\Delta_{t+1} = A_{1,t+1}\Delta_1 + \sum_{i=1}^{t-1} w_{i+1,t+1}\xi_{i+1} + \frac{\xi_{t+1}}{t+1}$$

$$= A_{1,t+1}\Delta_1 + \sum_{i=1}^{t} w_{i+1,t+1}\xi_{i+1},$$

which finishes the induction step.

To prove the lower bound expressed in Equation 24, consider $\beta = \sup_{r\in[0,1]} f'(r)$. Since $f(\mu^*) = \mu^*$, and $f$ is differentiable on the compact interval $[0,1]$, by the mean-value theorem

$$\frac{f(\mu) - \mu^*}{\mu - \mu^*} = \frac{f(\mu) - f(\mu^*)}{\mu - \mu^*} \in \{f'(z) : z \text{ between } \mu \text{ and } \mu^*\} \subseteq \left(-\infty, \beta\right].$$

Therefore, we have the following lower bound for $\alpha_t$ that holds for every $t \in \mathbb{N}$:

$$\alpha_t = 1 - \frac{f(\mu) - \mu^*}{\mu - \mu^*} \geq 1 - \beta. \tag{26}$$

For $u \in (0,1)$, we can use the bound $\log(1-u) \leq -u$. We can apply this to the definition of $A_{i,t}$

$$\log A_{i,t} = \sum_{j=i}^{t-1} \log\left(1 - \frac{\alpha_j}{j+1}\right) \leq -\sum_{j=i}^{t-1} \frac{\alpha_j}{j+1} \leq (\beta - 1)\log\frac{t+1}{i+1} = \log\left(\frac{i+1}{t+1}\right)^{1-\beta}. \tag{27}$$

The last inequality stems from using twice that $\sum_{i=1}^{t} 1/t \leq \log(t+1)$. Using $(i+1)/i \leq 2$, we have

$$w_{i,t}^2 = \left(\frac{A_{i,t}}{i}\right)^2 \leq \frac{1}{i^2}\left(\frac{i+1}{t+1}\right)^{2-2\beta} \leq \left(\frac{2}{t+1}\right)^{2-2\beta} i^{-2\beta}.$$

Using this bound, we have

$$\sum_{i=1}^{t-1} w_{i+1,t}^2 \leq \frac{4^{1-\beta}}{(t+1)^{2-2\beta}}\sum_{i=1}^{t-1}(i+1)^{-2\beta} \leq_{(*)} \frac{4^{1-\beta}}{(t+1)^{2-2\beta}}(t-1)t^{-2\beta} \leq \frac{4^{1-\beta}}{t+1},$$

where in the $(*)$ inequality we are bounding each element of the sum by tha largest one, which is $t^{-2\beta}$, and that there are $t-1$ summands. $\qquad\square$

**Lemma 5.** *Let $\delta > 0$. Let $\tau = \frac{2^{1+\frac{1}{1-\beta}}}{\delta^{1-\beta}} - 1$, and $K = (1/32) \cdot 4^\beta$. For all $t \geq \tau$, we have*

$$\mathbb{P}[\Delta_t > \delta] \leq \exp\left(-K\delta^2 t\right), \qquad \mathbb{P}[\Delta_t < -\delta] \leq \exp\left(-K\delta^2 t\right). \tag{28}$$

*Proof.* The idea is to use the expression in Equation 22 to bound $\Delta_t$. Of the two summands, $A_{1,t}\Delta_1$ will be bounded by bounding $A_{1,t}$ for large enough $t$. For the second summand, we will show it is a martingale and use a concentration inequality to bound its value. We show the case for the first inequality. The second inequality follows the same argument, with a symmetric use of the Azuma-Hoeffding's inequality.

Fix $t \in \mathbb{N}$ and consider $E_{i,t} = \sum_{j=1}^{i} w_{i+1,t}\xi_{i+1}$, for $i < t$. When conditioning over the decisions up to time $i$, we have $\mathbb{E}[\xi_{i+1} \mid \mu_i] = 0$, and therefore

$$\mathbb{E}[E_{i,t} \mid \mu_i] = E_{i-1,t} + w_{i+1,t}\mathbb{E}[\xi_{i+1} \mid \mu_i] = E_{i-1,t}.$$

So $(E_{i,t})_{i=1}^{t-1}$ is a martingale. Its increments are bounded by $|E_{i,t} - E_{i-1,t}| = |w_{i+1,t}\xi_{i+1}| \leq |w_{i+1,t}|$. Applying Azuma-Hoeffding's inequality to $E_{t-1,t}$ we have, for any $\delta > 0$

$$\mathbb{P}[E_{t-1,t} > \delta] \leq \exp\left(-\frac{\delta^2}{2\sum_{i=1}^{t-1} w_{i+1,t}^2}\right) \leq \exp\left(-\frac{(t+1)\delta^2}{2 \cdot 4^{1-\beta}}\right) \tag{29}$$

From Equation 27, applied to $i = 1$, we have

$$A_{1,t} \leq \left(\frac{2}{t+1}\right)^{1-\beta}.$$

We can apply the triangle inequality to Equation 22 to get

$$|\Delta_t| \leq |A_{1,t}\Delta_1| + |M_{t-1,t}|. \tag{30}$$

So we can apply the bound in Equation 29 with $\delta/2$ to bound $\mathbb{P}[\Delta_t > \delta]$ as long as $t$ is large enough to guarantee $|A_{1,t}\Delta_1| \leq \delta/2$. In such case

$$\mathbb{P}[\Delta_t > \delta] \leq \mathbb{P}[E_{t-1,t} > \delta/2] \leq \exp\left(-\frac{(t+1)(\delta/2)^2}{2 \cdot 4^{1-\beta}}\right) = \exp\left(-K(t+1)\delta^2\right), \tag{31}$$

with $K = (8 \cdot 4^{1-\beta})^{-1} = (1/32) \cdot 4^\beta$. For the previous bound to hold, we need $t$ large enough so that $|A_{1,t}\Delta_1| \leq \delta/2$. Using $|\Delta_1| \leq 1$ and Equation 30, we have

$$|A_{1,t}\Delta_1| \leq \left(\frac{2}{t+1}\right)^{1-\beta}, \tag{32}$$

which is smaller than $\delta/2$ for $t \geq \frac{2^{1+\frac{1}{1-\beta}}}{\delta^{1-\beta}} - 1$. $\qquad\square$

**Theorem 1.** *Let $I = [L, U]$ such that $\kappa, p \in I$. Let $\tau = 4/\min(|L - \mu^*|, |U - \mu^*|)$. Then for every $t \geq \tau$ we have*

$$\mathbb{P}(M_t \notin I) \leq \exp\left(-Kt|L - \mu^*|^2\right) + \exp\left(-Kt|U - \mu^*|^2\right), \tag{33}$$

*where $K$ is a positive constant defined as $K = (1/32) \cdot 4^\beta$ and $\beta = \sup_{r \in [0,1]} f'(r)$.*

*Proof.* This is just a matter of unpacking the results from Lemma 5 into the case of having a concrete interval. First note that

$$\mathbb{P}[M_t \notin I] = \mathbb{P}[M_t < L] + \mathbb{P}[M_t > U] = \mathbb{P}\left[\Delta_t < -|L - \mu^*|\right] + \mathbb{P}\left[\Delta_t > |U - \mu^*|\right].$$

We bound each of the summands using Lemma 5, which is guaranteed to hold as long as $t \geq \frac{2^{1+\frac{1}{1-\beta}}}{\delta^{1-\beta}} - 1$. Since $\beta \leq 0$, we have that

$$\frac{2^{1+\frac{1}{1-\beta}}}{\delta^{1-\beta}} - 1 \leq \frac{4}{\delta} - 1 \leq 4/\delta,$$

so if $t \geq 4/\delta$, Lemma 5 holds. This is exactly the condition that $t \geq \tau$, for $\tau = 4/\min(|L - \mu^*|, |U - \mu^*|)$. $\qquad\square$

**Corollary 1.** *Let* $r_- = \exp(-K|L - \mu^*|^2)$, $r_+ = \exp(-K|U - \mu^*|^2)$. *For every* $T, T' \in \mathbb{N}$ *such that* $\tau \le T < T'$ *we have:*

$$\mathcal{E}_\mathcal{S}(M_{T:T'}) \le \sum_{t=T}^{T'} (r_-^t + r_+^t) \qquad and \qquad \mathcal{E}_\mathcal{S}(M_{T:\infty}) \le \frac{r_-^T}{1 - r_-} + \frac{r_+^T}{1 - r_+}.$$

*Moreover, this provides the upper bound* $\mathcal{P}_\mathcal{S}(M_{T:T'}) \le \mathcal{E}_\mathcal{S}(M_{T:T'})$ *for* $T' \in \mathbb{N} \cup \{\infty\}$ *s.t.* $T' \ge T$.

*Proof.* This corollary is just an application of Boole's inequality (also known as union bound) on the results from Theorem 1. $\qquad\square$

### A.3 SHORT-TERM GUARANTEES: MONOTONICITY

To prove Theorem 2, we will use an inductive argument and the tower property of the expectation.

**Definition 4** (Tower operator). *Let* $\mathcal{U} = \{u \colon [0, 1] \to \mathbb{R}\}$ *be the space of all measurable functions from* $[0, 1]$ *to* $\mathbb{R}$. *Let* $t \in \mathbb{N}$, *and* $\zeta$ *be an energy function. The* towering operator *is the function* $T_{\zeta,t} \colon \mathcal{U} \to \mathcal{U}$, *that takes a function* $u$ *and returns* $T_{\zeta,t}u$ *defined as*

$$(T_{\zeta,t}u)(y) := f_\zeta(y)\, u\big(y_t^+(y)\big) + \big(1 - f_\zeta(y)\big)\, u\big(y_t^-(y)\big),$$

*where* $y_t^+(y) := y + \frac{1-y}{t+1}$ *and* $y_t^-(y) := y - \frac{y}{t+1}$.

*The* iterated tower operator *is* $T_\zeta^{(t)} = T_{\zeta,1} \circ T_{\zeta,2} \circ \cdots \circ T_{\zeta,t}$

**Lemma 6.** *Let* $u, v \colon [0, 1] \to \mathbb{R}$, $y \in [0, 1]$, $t \in \mathbb{N}$, $\zeta$ *an energy function and* $(y_k)$ *be a stochastic process generated following Eq. equation 3. The following properties hold:*

- Expectation: $\mathbb{E}_\zeta[u(y_t)] = \mathbb{E}_\zeta[T_{\zeta,t-1}u(y_{t-1})]$

- Iterated expectation: $\mathbb{E}_\zeta[u(y_t)] = \mathbb{E}\left[T_\zeta^{(t-1)}u(y_1)\right]$.

- Monotonicity: *If* $u(y) \le v(y)$, *then* $(T_{\zeta,t}u)(y) \le (T_{\zeta,t}v)(y)$.

*Proof.* The first property follows from a simple computation of the expectation, using that $\mathbb{P}[y_{t+1} = y_t^+ \mid y_t] = f_\zeta(y_t)$. In fact, the tower operator is defined with the expectation property in mind. Applying the expectation property consecutively leads to the next property

$$\mathbb{E}_\zeta[u(y_t)] = \mathbb{E}_\zeta[T_\zeta^{(t-1)}u(y_1)].$$

Monotonicity follows directly from the definition, noting that both $f_\zeta(y^+)$ and $1 - f_\zeta(y^-)$ are always non-negative. $\qquad\square$

The iterated expectation property is useful because it lets us write the expectation of $u(y_m)$, which depends on the distribution of $y_m$ after $m$ steps of the process, in terms of the expectation of a function depending only on the much simpler distribution of $y_1$. In particular, the distribution over $y_1$ does not depend on $\zeta$. If we want to prove a result of the form $\mathbb{E}_{\zeta_1}[u(y_t)] \le \mathbb{E}_{\zeta_2}[u(y_t)]$ for some $u, \zeta_1$, and $\zeta_2$, it is equivalent to prove that $\mathbb{E}[T_{\zeta_1}^{(t-1)}u(y_1)] \le \mathbb{E}[T_{\zeta_2}^{(t-1)}u(y_1)]$, so it suffices to prove that

$$T_{\zeta_1}^{(t-1)}u(y) \le T_{\zeta_2}^{(t-1)}u(y), \qquad \text{for} \quad y \in \{0, 1\}. \tag{34}$$

This observation lets us go from a local comparison to a global result. We study two properties on the short term: the probability of violating point fairness, and the expected number of point fairness violations. For the first one, we can take $u(y) = \mathbf{1}\{y \notin (L, 1 - U\}$. For the second one, we can take $u(y) = (L - y_k)_+ + (y_k - (1 - U))_+$.

**Lemma 7.** *Consider the functions* $y_t^+, y_t^-$ *as defined in Def. 4. Then*

1. *If* $y \le \kappa$ *(start below):*

   (a) *If* $y_t^+(y) \le \kappa$ *(no overshoot), then* $|y_t^+(y) - \kappa| < |y_t^-(y) - \kappa|$.

(b) If $y_t^+(y) > \kappa$ (overshoot), then

$$|y_t^+(y) - \kappa| \leq \frac{1 - \kappa}{t + 1}, \qquad |y_t^-(y) - \kappa| \leq \frac{1}{t + 1}. \qquad (35)$$

2. If $y > \kappa$ (start above):

(a) If $y_t^-(y) \geq \kappa$ (no overshoot), then $|y_t^+(y) - \kappa| > |y_t^-(y) - \kappa|$,

(b) If $y_t^-(y) < \kappa$ (overshoot), then

$$|y_t^-(y) - \kappa| \leq \frac{\kappa}{t + 1}, \qquad |y_t^+(y) - \kappa| \leq \frac{1}{t + 1}. \qquad (36)$$

*Proof.* We follow the proof case by case.

- *Case 1(a).* $|y_t^-(y) - \kappa| - |y_t^+(y) - \kappa| = y_t^+(y) - y_t^-(y) = 1/(t+1) > 0.$

- *Case 1(b)(left).*

$$|y_t^+(y) - \kappa| = \frac{1 - y}{t + 1} - (\kappa - y) = \frac{1 - \kappa t - \kappa + yt}{t + 1}.$$

Using $y \leq \kappa$ in the previous equation, we can cancel $-\kappa t$ with $\kappa t$, and get the result.

- *Case 1(b)(right).* Using $y_t^-(y) \geq 0$, we have $|y_t^-(y) - \kappa| = \kappa - y_t^-(y) < \kappa.$

- *Case 2(a).* $|y_t^+(y) - \kappa| - |y_t^-(y) - \kappa| = y_t^+(y) - y_t^-(y) = 1/(t+1) > 0.$

- *Case 2(b)(left).*

$$|y_t^-(y) - \kappa| = \frac{y}{t + 1} - (y - \kappa) = \frac{y - yt - y + \kappa t + \kappa}{t + 1}.$$

We can cancel $y$ and $-y$, and using $\kappa \leq y$, we can cancel $yt$ with $-yt$, and get the result.

- *Case 2(b)(right).* Simply note that $|y_t^+(y) - \kappa| \leq |y_t^+(y) - y_t^-(y)| = 1/(t+1).$

$\square$

**Lemma 8.** *Let $\zeta_1 \succeq \zeta_2$. Let $T \in \mathbb{N}$. Let $u$ be a unimodal function, minimized at $\kappa$, and with $u(y) = 0$ for $y \in [a, b]$, where*

$$a \leq \kappa - \frac{1}{T + 1}, \qquad b \geq \kappa + \frac{1}{T + 1}.$$

*Then for all $t \geq T$, we have for $y \in \{0, 1\}$ that*

$$T_{\zeta_1}^{(t)} u(y) \leq T_{\zeta_2}^{(t)} u(y).$$

*Proof. Step 1: a pointwise one-step inequality.* We first prove that for any $t \in \mathbb{N}$, $y \in [0, 1]$, and any measurable function $v$, symmetric around $\kappa$, minimized at $\kappa$ and null on $[a, b]$, we have

$$(T_{\zeta_1, t} v)(y) \leq (T_{\zeta_2, t} v)(y). \qquad (37)$$

Using the definition of the tower operator, we have

$$(T_{\zeta_1, t} v)(y) - (T_{\zeta_2, t} v)(y) = \left( f_{\zeta_1}(y) v(y_t^+) + (1 - f_{\zeta_1}(y)) v(y_t^-) \right) - \left( f_{\zeta_2}(y) v(y_t^+) + (1 - f_{\zeta_2}(y)) v(y_t^-) \right)$$

$$= f_{\zeta_1}(y) \left( v(y_t^+) - v(y_t^-) \right) - f_{\zeta_2}(y) \left( v(y_t^+) - v(y_t^-) \right)$$

$$= \left( f_{\zeta_1}(y) - f_{\zeta_2}(y) \right) \cdot \left( v(y_t^+) - v(y_t^-) \right).$$

Therefore, to check $(T_{\zeta_1, t} v)(y) - (T_{\zeta_2, t} v)(y) \leq 0$ we just need to prove that both factors of the previous equation are of different sign, or one of them is zero.

We check both sides of $\kappa$ separatedly.

Case $y \leq \kappa$. Since $\zeta_1 \succeq \zeta_2$, we have $f_{\zeta_1}(y) \geq f_{\zeta_2}(y)$. If we are in the no-overshoot regime $(y_t^+(y) \leq \kappa)$, we have $v$ in its descending mode in the whole interval $[y_t^-(y), y_t^+(y)]$, so $v(y_t^+(y)) \leq v(y_t^-(y))$. Therefore $f_{\zeta_1}(y) - f_{\zeta_2}(y)$ and $v(y_t^+) - v(y_t^-)$ have opposed signs.

If we are in the overshoot regime $(y_t^+(y) \leq \kappa)$, we want to make sure that $v(y_t^+) = v(y_t^-) = 0$. Using Eq. equation 35, we can guarantee it with

$$a \leq \kappa - \frac{1}{t+1}, \qquad \text{and} \qquad b \geq \kappa + \frac{1-\kappa}{t+1},$$

which is guaranteed for $t \geq T$ in the hypothesis of the lemma.

Case $y > \kappa$. This case is analogous, following the same argument as in $y \leq \kappa$, taking into account the switch in signs.

*Step 2: iterate the one-step inequality.* For $r = 0, 1, \ldots, t$ define

$$\Phi_r := B_1 B_2 \cdots B_r \, A_{r+1} A_{r+2} \cdots A_t \, u,$$

with the conventions

$$A_s := T_{\zeta_1, s}, \qquad B_s := T_{\zeta_2, s}, \qquad \Phi_0 = A_1 A_2 \cdots A_t u = T_{\zeta_1}^{(t)} u, \qquad \Phi_t = B_1 B_2 \cdots B_t u = T_{\zeta_2}^{(t)} u.$$

Our goal is to show the chain of inequalities

$$\Phi_0(y) \leq \Phi_1(y) \leq \cdots \leq \Phi_t(y), \qquad \text{for } y \in \{0, 1\}, \tag{38}$$

which implies $T_{\zeta_1}^{(t)} u(y) \leq T_{\zeta_2}^{(t)} u(y)$ at the endpoints.

*Telescoping principle.* For each $r = 1, \ldots, t$, set

$$v^{(r)} := A_{r+1} A_{r+2} \cdots A_t u.$$

Then

$$\Phi_{r-1} = B_1 \cdots B_{r-1} \left( A_r v^{(r)} \right), \qquad \Phi_r = B_1 \cdots B_{r-1} \left( B_r v^{(r)} \right).$$

Thus, if we can show

$$A_r v^{(r)}(y) \leq B_r v^{(r)}(y), \qquad y \in \{0, 1\}, \tag{39}$$

then applying the prefix operator $B_1 \cdots B_{r-1}$ (which is order-preserving because of the monotonicity property in Lemmas 6) yields

$$\Phi_{r-1}(y) \leq \Phi_r(y) \qquad \text{for } y \in \{0, 1\}.$$

Chaining over $r = 1, \ldots, t$ establishes the desired inequality.

*Verification of equation 39.* Fix $r$ and write $y^+ := y_r^+(y)$, $y^- := y_r^-(y)$ for brevity. By the algebraic identity from Step 1,

$$(A_r v^{(r)} - B_r v^{(r)})(y) = (f_{\zeta_1}(y) - f_{\zeta_2}(y)) (v^{(r)}(y^+) - v^{(r)}(y^-)).$$

At the endpoints $y \in \{0, 1\}$, the sign of $f_{\zeta_1} - f_{\zeta_2}$ is fixed by the assumption $\zeta_1 \succeq \zeta_2$, while the sign of $v^{(r)}(y^+) - v^{(r)}(y^-)$ is determined by the geometry of the successors and the plateau assumption:

- For $y = 0$: we have $f_{\zeta_1}(0) \geq f_{\zeta_2}(0)$. If $y^+ \leq \kappa$ (no overshoot), then $|y^+ - \kappa| < |y^- - \kappa|$, hence $v^{(r)}(y^+) \leq v^{(r)}(y^-)$. If $y^+ > \kappa$ (overshoot), then by the plateau condition both $y^\pm \in [a, b]$, so $v^{(r)}(y^\pm) = 0$. In both cases, $v^{(r)}(y^+) - v^{(r)}(y^-) \leq 0$, so the product is $\leq 0$.

- For $y = 1$: we have $f_{\zeta_1}(1) \leq f_{\zeta_2}(1)$. If $y^- \geq \kappa$ (no overshoot), then $|y^- - \kappa| < |y^+ - \kappa|$, hence $v^{(r)}(y^-) \leq v^{(r)}(y^+)$, i.e. $v^{(r)}(y^+) - v^{(r)}(y^-) \geq 0$. If $y^- < \kappa$ (overshoot), then $y^\pm \in [a, b]$, so $v^{(r)}(y^\pm) = 0$. In both cases, $v^{(r)}(y^+) - v^{(r)}(y^-) \geq 0$, so the product is $\leq 0$.

Thus $(A_r v^{(r)} - B_r v^{(r)})(y) \leq 0$ for $y \in \{0, 1\}$, proving equation 39, which in turn proves Eq. equation 38, as we wanted. $\qquad \square$

**Theorem 2.** *Let $\zeta_1 \succeq \zeta_2$ be two energy functions, with a common minimum at $\kappa$. Let $\mathcal{S} = [L, U]$. Let $M^{\zeta_1}$ and $M^{\zeta_2}$ be the shielded fairness process generated by enforcing the decision process of $p \in [0, 1]$ with $\zeta_1$ and $\zeta_2$, respectively. Let $\tau = \lceil \max\{1/|\kappa - L|, 1/|\kappa - U|\}\rceil$, for all $T \in \mathbb{N}$ and $T' \in \mathbb{N} \cup \{\infty\}$ such that $T < T'$ we have*

$$\mathcal{E}_{\mathcal{S}}(M_{T:T'}^{\zeta_1}) \leq \mathcal{E}_{\mathcal{S}}(M_{T:T'}^{\zeta_2}), \qquad \text{and} \qquad \mathcal{P}_{\mathcal{S}}(M_{T:T'}^{\zeta_1}) \leq \mathcal{P}_{\mathcal{S}}(M_{T:T'}^{\zeta_2}).$$

*Proof.* This is a direct consequence of Lemma 8. The condition on $\tau$ follows from it, and the condition on the values of either probability or expectation of point fairness violation come from the characterization of both of the safety measures as the expectation with certain functions $u$. In particular, for the first one, we can take $u(y) = \mathbf{1}\{y \notin (L, U)\}$, and for the second one, we can take $u(y) = (L - y_k)_+ + (y_k - U)_+$. The expectation of the outcome at a certain timestep corresponds to the expectation of the iterated tower operator at the first timestep because of Lemma 6. $\qquad\square$

### A.4 SHIELD SYNTHESIS

**Markov chain construction.** We construct the acyclic Markov chain $\mathcal{M} = (S, P, s_0)$ consisting of a set of states $S$, a Markov transition kernel $P\colon S \times S \to [0, 1]$, and an initial state $s_0 = (0, 0)$. Intuitively, the set of states is stratified into time steps $S = \bigcup_{t \in [0;T]} S_t$ where $S_t = \{(t, m) \mid m \in [t]\}$ for all $t \in [0; T]$. Intuitively, a state $s = (t, c) \in S$ is labeled with a point in time $t$, the positive decision counter $c$ clearly upper bound by the point in time $t$. The Markov transition kernel $P$ is defined as follows: for every state $s = (t, c)$ with $t \in [T - 1]$ we transition to state $s = (t + 1, c + 1)$ with probability $f(c/t)$, and to state $s' = (t + 1, c)$ with probability $1 - f(c/t)$.

**Dynamic programming.** We compute the value of both the probabilistic and expected violation measure for the interval $[1; T]$ using dynamic programming on the Markov chain $(S, P, s_0)$. Let $\gamma(s) = \mathbf{1}[c/t \notin \mathcal{S}]$, then for every $s = (t, c) \in S$ we define its value $V_{\mathcal{E}}$ for the expected violation measure as

$$V_{\mathcal{E}}(s) = \gamma(s) + (f(c/t)V_{\mathcal{E}}(t + 1, c + 1) + (1 - f(c/t))V_{\mathcal{E}}(t + 1, c)) \qquad \text{if } t \in [T - 1]$$
$$V_{\mathcal{E}}(s) = \gamma(s) \qquad\qquad\qquad\qquad\qquad\qquad\qquad\qquad\qquad\qquad\quad \text{if } t = T$$

Moreover, we define its value $V_{\mathcal{P}}$ for the expected violation measure as

$$V_{\mathcal{P}}(s) = \max\Big(\gamma(s), f(c/t)V_{\mathcal{P}}(t + 1, c + 1) + (1 - f(c/t))V_{\mathcal{P}}(t + 1, c)\Big) \qquad \text{if } t \in [T - 1]$$
$$V_{\mathcal{P}}(s) = \gamma(s) \qquad\qquad\qquad\qquad\qquad\qquad\qquad\qquad\qquad\qquad\qquad\quad \text{if } t = T$$

Notice that if $\gamma(c, t) = 1$ no further computation recursion is required.

**Intuition.** The value function $V_{\mathcal{E}}(s)$ represents the expected number of violations incurred from state $s = (t, c)$ onward, including a possible violation at time $t$. Formally, our process $M$ takes on the value $M_t = c/t$ at time $t$, then

$$V_{\mathcal{E}}(t, c) = \mathbf{1}[M_t \notin \mathcal{S}] + \mathbb{E}\left[\sum_{i=t+1}^{T} \mathbf{1}[M_i \notin \mathcal{S}] \,\middle|\, M_t\right].$$

Hence, the value at the initial state $s_0 = (0, 0)$ equals the total expected number of violations up to time $T$:

$$V_{\mathcal{E}}(s_0) = \mathbb{E}\left[\sum_{i=1}^{T} \mathbf{1}[M_i \notin \rho(i)]\right].$$

The value function $V_{\mathcal{P}}(s)$ instead captures the probability of encountering at least one violation from state $s$ onward. This probability equals 1 if $M_t \notin \mathcal{S}$, and otherwise it coincides with the probability of observing a violation at some later time:

$$V_{\mathcal{P}}(t, c) = \mathbb{P}\left[\max_{i \in [t;T]} \mathbf{1}[M_i \notin \mathcal{S}] = 1 \,\middle|\, M_t\right].$$

In particular, $V_{\mathcal{P}}(s_0)$ gives the probability of observing any violation within the time horizon $[1, T]$.

## B    FAMILIES OF ENERGY FUNCTIONS OF INTEREST

In this section we propose two families of general-purpose energy shields, and one family of energy shields specifically designed to fit a given specification. Note that some of these functions have isolated non-smooth points, where the function is continuous but not differentiable. These do not pose a theoretical issue, since we can always "glue" the non-smooth endpoints of the two smooth pieces using infinitely differentiable bump functions[1].

### B.1    POLYNOMIAL

$$\zeta_{\kappa,\alpha,\beta}^{Pol}(x) = \alpha|x - \kappa|^{\beta},$$

where $\beta \in (1, \infty)$, $\kappa \in (0,1)$ and $\alpha \in \left(0, \frac{1}{\max\{\kappa, 1-\kappa\}^{\beta}}\right)$. In this family, $\kappa$ marks the pivoting point, and $\alpha$ and $\beta$ control the shape, with larger $\alpha$ and $\beta$ producing steeper functions.

### B.2    EXPONENTIAL

$$\zeta_{\kappa,\alpha,\beta}^{Exp}(x) = \alpha\left(1 - e^{-\beta(x-\kappa)^2}\right),$$

where $\beta \in (0, \infty)$, $\kappa \in (0,1)$, and $\alpha \in \left(0, \frac{1}{1-e^{-\beta(\min\{\kappa, 1-\kappa\})^2}}\right)$.

### B.3    CONSTRUCTION OF A MONOTONIC FAMILY OF ENERGY FUNCTIONS

Let $p \in (0,1)$, $\varphi = (\tau, \mathcal{S}, \mathcal{L})$ be a specification, with $\mathcal{S} = [L_{\mathcal{S}}, U_{\mathcal{S}}]$ and $\mathcal{L} = [L_{\mathcal{L}}, U_{\mathcal{L}}]$, $\mathcal{L} \subset \mathcal{S}$. We give the construction for $\zeta_{r;p,\mathcal{S},\mathcal{L}}^{\text{Mon}}$. For ease of notation, within this section we will call it just $\zeta_r$. The family of monotonic function is built differently depending on the relative position of $p$ and $\mathcal{L}$.

- **If $p < L$.** The family of energy functions is $(\zeta_r)_{r \in R}$, with $R = (0,1)$, and is built as follows. Let $\kappa = (U_{\mathcal{L}} + U_{\mathcal{S}})/2$, $a_r = (1-r)L_{\mathcal{L}} + rU_{\mathcal{L}}$, $C_r = (a_r - p)/(1-p)$, $\alpha_r = (1-r)/r$.

$$\zeta_r(x) = \begin{cases} \zeta_r^1(x) & \text{if } x < a_r, \\ \zeta_r^2(x) & \text{if } a_r \leq x \leq \kappa, \\ \zeta_r^3(x) & \text{otherwise.} \end{cases} \tag{40}$$

    With the following definitions:

$$\zeta_r^1(x) = C_r + (1 - C_r)\left(1 - e^{\frac{x - a_r}{\alpha_r}}\right),$$

$$\zeta_r^2(x) = C_r\left(1 - \frac{x - a_r}{\kappa - a_r}\right)^{\alpha_r},$$

$$\zeta_r^3(x) = 1 - \exp\left(-\left(\frac{x - \kappa}{\alpha_r}\right)^m\right).$$

    where $m$ is a fixed integer $m \geq 2$, we choose $m = 2$.

- **If $p > U$.** We follow a symmetric construction as the previous case. The family of energy functions is $(\zeta_r)_{r \in R}$, with $R = (0,1)$, and is built as follows. Let $\kappa = (L_{\mathcal{S}} + L_{\mathcal{L}})/2$, $a_r = rL_{\mathcal{L}} + (1-r)U_{\mathcal{L}}$, $C_r = (p - a_r)/p$, $\alpha_r = (1-r)/r$.

$$\zeta_r(x) = \begin{cases} \zeta_r^1(x) & \text{if } x < \kappa, \\ \zeta_r^2(x) & \text{if } \kappa \leq x \leq a_r, \\ \zeta_r^3(x) & \text{otherwise.} \end{cases} \tag{41}$$

---

[1]This follows the same rationale as why most theoretical results in convergence of machine learning algorithms work with the non-smooth ReLU activation function.

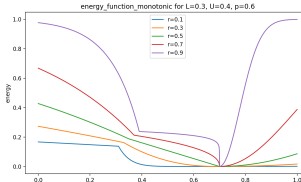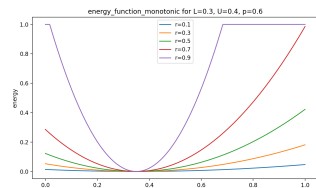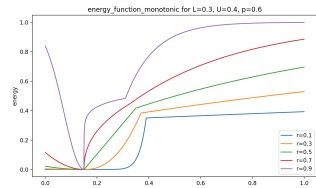

Figure 2: Monotonic families of functions

With the following definitions:

$$\zeta_r^1(x) = 1 - e^{-\left(\frac{x-\kappa}{\alpha_r}\right)^m}$$

$$\zeta_r^2(x) = C_r \left(1 - \frac{a_r - x}{a_r - \kappa}\right)^{\alpha_r},$$

$$\zeta_r^3(x) = C_r + (1 - C_r)\left(1 - e^{\frac{a_r - x}{\alpha_r}}\right).$$

- **If $p \in [L_\mathcal{L}, U_\mathcal{L}]$.** In this case, the natural bias of the process already aligns with the short-term requirements, so we can choose $\kappa = p$ and use a family of either polynomial or exponential functions. We show here a family of modified polynomial functions that satisfy the monotonicity requirements. The family of energy functions is $(\zeta_r)_{r \in R}$, with $R = (0, 1)$, built as follows. Let $\alpha_r = r/(1-r)$, and $m \geq 2$ fixed, $l_r = \kappa - \frac{1}{\alpha_r^{1/m}}$, $u_r = \kappa + \frac{1}{\alpha_r^{1/m}}$. Then

$$\zeta_r(x) = \begin{cases} \alpha_r |x - \kappa|^m & \text{if } x \in [l_r, u_r] \\ 1 & \text{otherwise.} \end{cases}$$

We use $m = 2$.

**Claim 2.** *The family of functions $(\zeta_r)_{r \in R}$ satisfies the following properties:*

- *It is monotonic in $r$, i.e., for each $x \in [0, 1]$ and each pair $s, r \in (0, 1)$, if $s \leq r$, then $\zeta_s(x) \leq \zeta_r(x)$.*

- *The corresponding characteristic function $f_r$ (as defined in Eq. 2) has a fixpoint at $\mu = a_r$.*

- *If $p < L_\mathcal{L}$, for all $x \neq \kappa$:*

$$\lim_{r \to 0} \zeta_r(x) = \begin{cases} (L_\mathcal{L} - p)/(1-p) & \text{if } x \leq L_\mathcal{L} \\ 0 & \text{otherwise.} \end{cases}, \quad \lim_{r \to 1} \zeta_r(x) = \begin{cases} (L_\mathcal{L} - p)/(1-p) & \text{if } x \in [U_\mathcal{L}, \kappa] \\ 1 & \text{otherwise.} \end{cases}$$

- *If $p > U_\mathcal{L}$, for all $x \neq \kappa$:*

$$\lim_{r \to 0} \zeta_r(x) = \begin{cases} (p - L_\mathcal{L})/p & \text{if } x \geq U_\mathcal{L} \\ 0 & \text{otherwise.} \end{cases}, \quad \lim_{r \to 1} \zeta_r(x) = \begin{cases} (p - L_\mathcal{L})/p & \text{if } x \in [\kappa, L_\mathcal{L}] \\ 1 & \text{otherwise.} \end{cases}$$

- *If $p \in \mathcal{L}$, for all $x \neq \kappa$ and all $l \in \{0, 1\}$, $\lim_{r \to l} \zeta_r(x) = l$.*

All the properties are satisfied by construction. Fig. 2 illustrates them.

