# OpenReview forum: "Energy Shields for Fairness"
_ICLR.cc/2026/Conference — ICLR 2026 Conference Withdrawn Submission_

### Official Review · Reviewer_yqGE · 2025-10-28

**Soundness:** 3
**Presentation:** 3
**Contribution:** 2
**Rating:** 4
**Confidence:** 4

**Summary:**

This paper proposes energy shields, a probabilistic, physics-inspired framework that enforces fairness at runtime by assigning higher “energy” to unfair decision trajectories and intervening with a probability proportional to that energy. Unlike prior deterministic fairness shields, the proposed approach offers both short-term (running) and long-term (limit) fairness guarantees. The authors provide theoretical results establishing exponentially decaying tail bounds for running fairness, convergence conditions for limit fairness, and a monotonicity property showing that steeper energy functions yield fewer violations. Leveraging this, they design a synthesis algorithm combining binary search and dynamic programming to derive the least intrusive shield satisfying both fairness requirements. Experimental results validate the practicality of the approach and show its advantages over previous fairness shields

**Strengths:**

1- The energy shield is the first probabilistic fairness controller capable of providing both running and limit fairness guarantees within one framework.

2- Offers provable probabilistic guarantees via exponentially decaying bounds on fairness violations and convergence proofs for long-term fairness stability.

3- Establishes that steeper energy functions reduce violations, forming the basis for an elegant synthesis algorithm using binary search and dynamic programming.

4- Balances fairness control and intervention cost, presenting a clear optimization objective for runtime fairness enforcement.

5- Connects algorithmic fairness with energy minimization principles and control-theoretic intuition, opening a new interdisciplinary perspective.

**Weaknesses:**

1- The decision process assumes a binary Bernoulli model with stationary probabilities, which limits realism for multi-class, contextual, or adaptive systems.

2- The experiments confirm basic properties but fail to explore scalability, robustness under non-stationarity, or performance on real-world datasets.

3- Theoretical analysis assumes specific functional forms (e.g., polynomial or exponential), yet no discussion is provided on how to choose or tune these for practical tasks.

4- The framework defines fairness in abstract mathematical terms; mapping it to standard group metrics (e.g., equal opportunity, demographic parity) is not demonstrated.

5- While rooted in formal verification, the paper does not clearly show how these results could integrate with modern fairness-aware learning or causal inference models.

**Questions:**

1- How can the proposed shielding mechanism be extended to non-binary or continuous decisions common in ML systems?

2- Can the energy function ζ be learned adaptively or parameterized using observed violations rather than predefined analytic forms?

3- Does the monotonicity property hold under model misspecification or non-stationary distributions?

4- How would the fairness targets (τ, S, L) map to conventional fairness metrics such as Equalized Odds or Demographic Parity in applied settings?

5- Can the proposed synthesis procedure scale to large decision horizons or online settings where dynamic fairness targets evolve over time?

6- Could the shield be integrated into reinforcement learning or policy optimization frameworks for sequential fairness?

---

> ### Author Response · Authors · 2025-11-22
> **Part 1**
>
> **W1.** As discussed in the general rebuttal, we agree that presenting only the setting with a known and fixed Bernoulli parameter $p$ can seem restrictive. However, our method extends to the settings (i)–(iii), which in particular include adaptive and multi-group scenarios. We will make these extensions explicit in the revised version by adding a dedicated discussion and corresponding experimental evaluations.
>
> **W2.** Our method is agnostic to the size of the decision maker and to the underlying data distribution. Once the shield is synthesised, it can be applied at runtime to traces of arbitrary length. This is a substantial improvement over [Cano et al., 2025b], where the shield construction scales polynomially with the horizon length. The main bottleneck in our shield synthesis is obtaining good approximations for the short-term guarantees, as illustrated in Fig. 1(b).
>
> **W3.** In our theoretical analysis, we intentionally kept the conditions on the energy function as general as possible. Any function satisfying these conditions induces a valid energy shield, regardless of its concrete functional form.
>
> To make this more tangible, we study several specific families of functions that satisfy the conditions in Appendix B. In Appendices B.1 and B.2, we propose parametric families of energy functions designed as general-purpose building blocks. In Appendix B.3, we show how to explicitly construct a monotone family of energy functions tailored to a given specification. Using similar principles, further monotone families can be designed.
>
> **W4.** Group fairness formalises equal treatment of demographic groups in binary decision settings and typically compares decision probabilities across groups. For example, demographic parity requires $\mathbb{P}(X = 1 \mid \text{Group} = A) - \mathbb{P}(X = 1 \mid \text{Group} = B)  \in [- \varepsilon,+\varepsilon]$ for some pre-specified small $\varepsilon \geq 0$, i.e., the acceptance probabilities are approximately equal.
>
> The shield in our paper is designed to control both the short-term and long-term behaviour of the empirical counterparts of such probabilities. For instance, if the shield operates on the sequence of decisions $X_1, \dots, X_t$ for Group A, then the Bernoulli parameter $p$ represents $\mathbb{P}(X = 1 \mid \text{Group} = A)$. Our shield ensures that the empirical acceptance rate $\mu(Z_1, \dots, Z_t)$ of the shielded process $Z_1, \dots, Z_t$ remains within a prescribed region with high probability in the short term and converges to a target value in the long term.
>
> This extends naturally to two groups. For demographic parity, let $\mu_G(Z_1, \dots, Z_t)$ be the empirical acceptance rate of the shielded process for group $G \in {A,B}$. Then we can design an energy shield that controls the difference $\mu_A(Z_1, \dots, Z_t) - \mu_B(Z_1, \dots, Z_t) $ by using an energy function defined over an extended domain and pushing this difference towards a desired target (e.g., zero). This construction is fleshed-out in more details in the revised version of the paper, Section 9.
>
> In the classical demographic parity view, the limit average of the decisions of a fair classifier converges to a value satisfying the demographic parity condition. In previous fairness shielding approaches [Cano et al. 2025b], the fairness guarantee is in the short-term. Our shields ensures that the fairness condition is satisfied both asymptotically and in the short term.
>
> Equalized Odds and Equal Opportunity are conceptually similar to demographic parity, and also reduce to ratios of the number of certain decisions for each group. The main issue with these measures, from the perspective of applying a sequential decision-making approach, is that they require a ground truth (e.g., whether a loan is actually repayed), which is not assumed to be available around the time that the shield decides whether to intervene or not.
>
> **W5.** As in our response to Review sSmq, we emphasise that shielding is complementary to pre-processing, in-processing, and post-processing approaches. Pre-deployment methods typically control fairness only in expectation. It is possible to calibrate a classifier to satisfy demographic parity during development, yet at runtime the realised empirical difference in acceptance rates may still be substantial in finite time. Our shield corrects this by ensuring that the empirical acceptance rates remain within a user-specified target region with high probability.

---

> ### Author Response · Authors · 2025-11-22
> **Part 2**
>
> **Q1.** Our fairness shield targets group fairness notions that are typically defined over binary decisions (e.g., a loan is either granted or rejected). In adaptive Bernoulli settings (as described by setting (ii) in the general rebuttal), we can interpret the parameter $p_t$ as a continuous decision probability from which a binary decision is derived either by sampling or by thresholding. Our framework directly applies to this setting.
>
> **Q2.** In principle, yes. The energy function encodes the desired behaviour of the shielded process. For the long-term target, any energy function with a fixpoint at the target value guarantees convergence to that value. Hence, we can adaptively choose the energy function to modulate the short-term behaviour while preserving the long-term fairness guarantees.
>
> In the revised version [Theorem 4] we also show how we can learn an energy function adaptatively in the setting where $p$ is unknown but fixed, so we can estimate it.
>
> **Q3.** In general, the monotonicity properties apply to a safety interval $[L, U]$ that contains the Bernoulli parameter $p$. Thus, we assume that $p$, even if unknown or evolving, always lies within $[L, U]$.
>
> * If the model is misspecified but fixed (i.e., $p$ is fixed but unknown), and the safety interval contains the true value of $p$, then the monotonicity property still holds, and the upper bound in Theorem 1 remains valid, with the caveat that $\mu^\ast$ is unknown.
> * If the model evolves in a way that is independent of the current trace (i.e., $p_t$ changes due to exogenous factors unaffected by the realised shielded decisions), the proof of the monotonicity result can be extended. In the proof of Lemma 8 (the main technical lemma behind Theorem 2), we assume that steeper energy functions correspond to certain inequalities for the characteristic function $f$ (defined in Eq. (2)), and we use that both processes are evaluated at the same value of $p$. If the parameters evolve over time but are always the same when comparing two processes driven by different energy functions, the inequalities and hence the monotonicity result continue to hold.
>
> **Q4.** See response **W4** above.
>
> **Q5.** For large horizons, the answer is essentially yes. The cost of *synthesising* the shield scales quadratically with the time horizon. This synthesis is required only if we wish to enforce short-term guarantees. For purely long-term guarantees, we can ensure convergence of the shielded process to the desired value by choosing an appropriate energy function in constant time.
>
> At runtime, there is no additional cost associated with increasing the horizon length. The current empirical average can be updated on the fly without storing the entire trace. As long as the counters fit in memory, the runtime cost of implementing energy shields is independent of the horizon length. This is a key advantage over the shields in [Cano et al., 2025b], where the horizon length is the main bottleneck for scalability.
>
> Our shields can also be adapted to a setting with changing bias.
> - If the bias evolves in an unknown way and we have to use the same energy function, the shielded process may not converge a.s., but we can show that a shield with a fixed energy function can still guarantee bounds for the limit inferior and limit superior. We explore this in the revised version (Section 8).
> - If the bias evolves in an unknown way and we can modify the energy function, we may estimate the bias as the empirical bias observed until that step, and use it to synthesize the energy function. Without further assumptions, this setting does not let us obtain any more convergence guarantees.
> - If the bias evolves in a way that we know its value at each time, we could modify the energy function every step to match a desired resulting bias. However, at this point, we are simply forcing the outcome to have our target probability at each time, so it is not very interesting.
>
>
> **Q6.** Our shields are agnostic to the underlying decision-making agent, so they can, in principle, be integrated into reinforcement learning frameworks. An interesting open question is how best to use the shield’s feedback to shape the reward signal for the learning agent. We expect that there is no universal answer and that the best strategy will be pplication-dependent; we leave this as future work.

---

### Official Review · Reviewer_couy · 2025-10-29

**Soundness:** 4
**Presentation:** 3
**Contribution:** 3
**Rating:** 6
**Confidence:** 2

**Summary:**

This paper presents a mechanism for synthesizing probabilistic energy shields for fairness in sequential decision-making. The setting is that there is a decision maker deciding true/false outcomes in a sequential fashion and one wants to enforce that after some burn-in time, some target of true/false decision ratios is maintained within some interval, and that in the limit the ratio converges to some given value. A shield is a monitor that occasionally nudges decisions to help the decision maker satisfy the fairness goal. The key gap to address is that deterministic shields can be "invasive/abrupt" and costly to execute as they require solving a dynamic program. The proposed approach is instead to have a probabilistic shield that only nudges with some probability, thus ensuring more "smooth" shielding. The key insight is to decide with what probability to nudge based on an energy function that captures how much the decisions are deviating from the target fairness goals. The results on two examples illustrate that the probabilistic controller achieves the desired goals compared to a deterministic one

**Strengths:**

- Lifting deterministic shields to probabilistic ones is a very natural direction
- The paper is technically well-executed and the theoretical contributions seem non-trivial (note, I'm not an expert here, so I can't really check how hard/novel this part is). It provides a comprehensive theoretical analysis of the properties of the algorithm.

**Weaknesses:**

- The evaluation is a bit underwhelming. The paper evaluates on 3 very similar artificial benchmarks and mostly considers a very naive decision makers that just samples from p. I'm not sure what the benchmarks are for this setting.
- The evaluation discusses that that other shields over-intervene or are invasive. I couldn't find a definition of these concepts in a way that is measurable. While I see that the plots illustrate some phenomenon of this sort, is there a way to actually compute a metric?

**Questions:**

The notion of fairness in this paper seems more similar to calibration than fairness itself. Can the authors discuss how their definitions related to calibration? (e.g., this paper https://proceedings.mlr.press/v247/qiao24a/qiao24a.pdf)

Have you considered applying this approach to the predictions of auto-regressive language models (aka LLMs)? I can imagine a setting in which some set of words (maybe negative sentiments or known-GPT words like delve) should not be "overly used" and so they have a target probability. The shield would then monitor usage and occasionally "re-weigh" the LLM logits to ensure usage stays within target? In such a setting I would expect that your approach is more beneficial than deterministic or periodic monitors as those would make the output fluency (something measurable) worse.


MINOR/OTHER:
- No need to cite this, or do anything about it, but this line of work reminds me of older results in synthesizing controllers (and the idea of algorithmic improvisation). I can't find the paper that comes to mind on making sure a thermostat keeps temperatures within some bounds while in the limit reaching a desired temperature, but I found this maybe related paper (https://susmitjha.github.io/papers/iccps10.pdf). Perhaps the authors know these works, or if not, they might find this pointer useful

---

> ### Author Response · Authors · 2025-11-22
>
> **W1.** To the best of our knowledge, shielding of fairness properties is still underexplored, and there are no established benchmarks. As mentioned in the general rebuttal, we are expanding our experiments to benchmark our shields against the fairness shields from [Cano et al., 2025b] on their existing real-world datasets.
>
> **W2.** This is an excellent point. The cost of interventions for shields is commonly quantified by simply counting the number of interventions, which is why we adopted this measure in our experiments [we show this notion of cost in Fig.1c]. This choice reflects the formal verification literature, where the *severity* of an intervention is often not quantified or sometimes not even quantifiable.
>
> In our setting, however, we can quantify the *degree* of intervention, which opens the door to richer measures of invasiveness. One natural measure would be $
> \frac{1}{t} \sum_{i=1}^t (p_i - \hat{p}_i)^2$ ,i.e., the time-average of the squared deviation between the original decision probability $p_i$ and the shielded decision probability $\hat{p}_i$. This can also be restricted to time steps where an intervention occurs, giving a measure of the *average strength* of an intervention.
>
> We will add a discussion of these alternative notions of invasiveness and run an additional set of experiments comparing different measures.
>
> ---
>
> **Q1.** Thank you for pointing this out. We can see the following  connection to calibration.
>
> Consider, for example, a loan-underwriting setting with a single group (say, Group A). Let the target, e.g., the fixpoint $\mu^\star$, represent the probability that a member of Group A repays the loan.
> A classifier that is perfectly calibrated for Group A has an acceptance rate that converges to $\mu^\star$ in the long term. When we equip any classifier with our shield, the shielded acceptance rate will converge to the same long-term value and remain close to this target in the short term. Thus, our shield can be used to enforce calibration during deployment.
>
> **Q2.** We have not explored this direction yet, but we agree it is a promising idea. A higher-dimensional generalisation of energy shields appears to be a natural fit for such problems, and we thank the reviewer for the suggestion.
>
> **Q3.** Thank you for the pointer. Indeed, our long-term results rely on techniques from stochastic approximation, which are closely related to the deterministic approximation scheme you mention. We will clarify this connection in the revised version.

---

### Official Review · Reviewer_sSmq · 2025-10-31

**Soundness:** 3
**Presentation:** 4
**Contribution:** 3
**Rating:** 4
**Confidence:** 3

**Summary:**

This paper introduces "energy shields," a new probabilistic approach to ensure fairness in sequential decision-making tasks. Unlike traditional deterministic methods, energy shields gently intervene based on physics-inspired energy functions, correcting unfair decisions with higher probability as unfairness increases. The authors define two fairness objectives, short-term running fairness and long-term limit fairness, and provide theoretical guarantees for both. They also propose an efficient synthesis method to identify the minimally intrusive shield and demonstrate experimentally that energy shields outperform existing deterministic fairness interventions.

**Strengths:**

1.	The method wraps any binary decision stream with a probabilistic controller that flips the current action with a probability derived from a bowl-shaped energy function. This yields smooth, minimally invasive corrections that gently steer outcomes toward a target without retraining models or exposing internals, making it easy to bolt onto existing systems.
2.	The paper derives explicit finite-time tail bounds for the probability and expectation of short-term fairness violations, providing mathematically rigorous safety guarantees. This substantially improves the theoretical credibility and reliability of the approach.
3.	Broader linkage and scope. This paper discusses how single-stream shielding relates to group fairness checks (e.g., demographic parity can reduce to controlling one Bernoulli stream) and connects short- vs long-term fairness to standard safety–liveness notions.

**Weaknesses:**

1. The paper’s empirical evidence comes entirely from simulated binary decision sequences rather than real-world datasets. This work is primarily theoretical and does not utilize datasets containing sensitive information, thereby confirming that no external real-world datasets were employed.
2. A limitation is the narrow choice of baselines. The paper evaluates energy shields only against two methods, which are both runtime-shielding approaches rather than broader fairness techniques. This restricts external comparability and makes it harder to assess how the method would compare to standard pre-processing, in-processing, post-processing, or other online controllers.
3. The authors may benefit from discussing how their runtime fairness-shielding method relates to existing online fairness-aware learning frameworks that explicitly optimize accuracy-fairness trade-offs during model training, such as [1] "FAHT: an adaptive fairness-aware decision tree classifier", and [2] "Preventing Discriminatory Decision-making in Evolving Data Streams".

**Questions:**

Please refer to the weaknesses

---

> ### Author Response · Authors · 2025-11-22
>
> **W1.** We originally omitted real-world datasets for the following reason. Given a real-world dataset and a classifier that makes a binary decision upon receiving a feature vector, we must choose a sampling distribution over the dataset (e.g., uniform over data points) to generate a stream. This induces a fixed acceptance probability obtained by taking the classifier’s expected decisions under that sampling distribution. This fixed acceptance probability is precisely our Bernoulli parameter $p$. Hence, for our purposes it is immaterial whether $p$ is derived from a real-world dataset or chosen synthetically.
>
> Note that this fixed acceptance rate (i.e., the fixed Bernoulli parameter $p$) must exist; otherwise, group fairness notions such as demographic parity are not well defined.
>
> As mentioned in the general rebuttal, we are now extending our experiments to the real-world datasets used in [Cano et al., 2025b], in the setting of demographic parity.
>
> **W2.** Shielding approaches are complementary to pre-processing, in-processing, and post-processing methods. These pre-deployment methods are typically limited to controlling fairness in expectation. For instance, a classifier can be calibrated to satisfy demographic parity at development time, while at runtime the actual difference in empirical acceptance rates, which is the quantity that actually matters in many real scenarios, may still be too large, particularly in the short term. Our shield directly controls the empirical acceptance rate, ensuring it remains within the desired target region over time.
>
> As part of the new round of experiments, we are shielding decision makers trained on learning algorithms with design-time fairness enforcing mechanisms, such as HSIC [a], LAFTR [b], and DiffDP [c].
>
> **W3.** Thank you for the helpful pointers. We will add a discussion placing our work in the context of the two suggested papers and clarifying the relationship between these approaches and energy shields.
>
> Both [1] and [2] study fairness in data streams, but they intervene at the *learning* level, whereas our work intervenes at *decision* time. [1] extends the Hoeffding Tree algorithm with a fairness-aware splitting criterion so that the learned tree remains fair as the stream evolves, and [2] proposes a fair online data rebalancing strategy. In contrast, our energy shields treat the underlying decision maker as a black box and operate purely at runtime, directly constraining the empirical acceptance rates (and their differences across groups) via probabilistic short-term and long-term guarantees. As such, our approach is complementary [1] and [2] can be used to learn fairer base models, while energy shields can be wrapped around any such model to further enforce fairness constraints on the realised decision stream.
>
> **References**
>
> - [a] Perez-Suay, A. et al. *Fair Kernel Learning*. KDD 2017.
> - [c] Madras, D. et al. *Learning adversarially fair and transferable representations*. ICML 2018.
> - [b] Chuang, C.; and Mroueh, Y.. *Fair Mixup: Fairness via Interpolation*. ICLR 2021.

---

### Author Response · Authors · 2025-11-22
**Proposed revisions**

## General Rebuttal

We thank the reviewers for their constructive feedback. We acknowledge that our presentation centred around the setting with a known and fixed Bernoulli parameter $p$, which may appear limiting. However, this setting already captures much of the technical difficulty of more general scenarios, which is why we focused on it in the main paper. Below, we outline how our method extends to the following settings:

- (i) fixed but unknown parameter $p$;
- (ii) varying and unknown parameter $p_t$;
- (iii) two fixed parameters $p_A$ and $p_B$ with quantity of interest $p_A - p_B$. At each time, an element is drawn from the population corresponding to $p_A$ or $p_B$ with a known probability $\pi$. This is the typical setting for modelling group fairness, where the population is divided into two groups of interest, sa groups $A$ and $B$, that should receive equal treatment.

In the current revised version of the paper we already formalized the above scenarios and demonstrated that the generalized shield satisfies the long-term guarantees for each scenario (sections 8 and 9 of the revised version).
We are currently working on writing up the short-term guarantees. Moreover, we will extend the experimental section to support this theoretical discussion:

We will implement experiments illustrating the behaviour of the shield in scenarios (i) and  (ii).
 We will implement scenario (iii) and benchmark our shield against the fairness shield from “Fairness Shields: Safeguarding against Biased Decision Makers” [Cano et al., 2025b] on their real-world datasets.

For an eventual final version, in order to respect the 10 page limit, we may move some of these scenarios to the appendix, or alternatively, shorten the examples and proof intuitions in Sections 4, 5, and 6.

---

### Note · Authors · 2025-12-03

**Comment:**

We believe that the paper will benefit from additional time to improve the overall presentation.
Hence, we decide to withdraw the paper.
Thank you for your consideration and effort.

**Withdrawal Confirmation:**

I have read and agree with the venue's withdrawal policy on behalf of myself and my co-authors.